

# High-resolution stable isotope signature of a land-falling Atmospheric River in southern Norway

Yongbiao Weng, Harald Sodemann, and Aina Johannessen

Geophysical Institute, University of Bergen, and Bjerknes Centre for Climate Research, Bergen, Norway

**Correspondence:** Yongbiao Weng (yongbiao.weng@uib.no) and Harald Sodemann (harald.sodemann@uib.no)

**Abstract.** Heavy precipitation at the west coast of Norway is often connected to elongated meridional structures of high integrated water vapour transport known as Atmospheric Rivers (AR). Here we present high-resolution measurements of stable isotopes in near-surface water vapour and precipitation during a land-falling AR event in southwestern Norway on 07 December 2016. In our analysis, we aim to identify the influences of moisture source conditions, weather system characteristics, and post-condensation processes on the isotopic signal in near-surface water vapour and precipitation.

A total of 71 precipitation samples were collected during the 24-h sampling period, mostly taken at sampling intervals of 10–20 min. The isotope composition of near-surface vapour was continuously monitored in-situ with a cavity ring-down spectrometer. Local meteorological conditions were in addition observed from a vertical pointing rain radar, a laser disdrometer, and automatic weather stations.

We observe a stretched, "W"-shaped evolution of isotope composition during the event. Combining isotopic and meteorological observations, we define four different stages of the event. The two most depletion periods in the isotope $\delta$ values are associated with frontal transitions, namely a combination of two warm fronts that follow each other within a few hours, and an upper-level cold front. The $d$-excess shows a single maximum, and a step-wise decline in precipitation and a gradual decrease in near-surface vapour. Thereby, isotopic evolution of the near-surface vapour closely follows the precipitation with a time delay of about 30 min, except for the first stage of the event. Analysis using an isotopic below-cloud exchange framework shows that the initial period of low and even negative $d$-excess in precipitation was caused by evaporation below cloud base. At the ground, a near-constant signal representative of the airmass above is only reached after transition periods of several hours. Moisture source diagnostics for the event show that the moisture source conditions for these steady periods are partly reflected in the surface precipitation at these times.

Based on our observations, we revisit the interpretation of precipitation isotope measurements during AR events in previous studies. Given that the isotopic signal in surface precipitation reflects a combination of atmospheric dynamics through moisture sources and atmospheric distillation, as well as cloud microphysics and below-cloud processes, we recommend caution regarding how Rayleigh distillation models are used during data interpretation. While the isotope compositions during convective precipitation events may be more adequately represented by idealized Rayleigh models, additional factors should be taken into account when interpreting a surface precipitation isotope signal from stratiform clouds.



# 1 Introduction

Precipitation can be considered as the end product of the atmospheric hydrological cycle. Weather systems lead to sequences of ocean evaporation, horizontal and vertical transport and mixing of atmospheric water vapour, microphysical processes within clouds on characteristic time scales (Läderach and Sodemann, 2016). The stable isotope composition of precipitation is,

therefore, an integrated result of the isotopic fractionation, that occurs during phase changes in the atmosphere (Gat, 1996). In addition, post-condensation processes can influence the isotope composition below cloud base (Graf et al., 2019). Therefore, observations of stable water isotopes in precipitation hold the promise of allowing to extract information about moisture transport and moisture sources for individual weather events. Besides, detailed measurements of water isotopes provide the potential to constrain parameterisations in atmospheric models and thereby to improve weather prediction and climate models

(Bony et al., 2008; Pfahl et al., 2012; Yoshimura et al., 2014).

Being located at the end of the North Atlantic storm track, precipitation on the west coast of Scandinavia is commonly related to the landfall of frontal weather systems. Extreme precipitation has been connected to so-called Atmospheric Rivers (ARs, Zhu and Newell, 1998; Ralph et al., 2004), that transport warm and moist air from more southerly latitudes poleward within their frontal structures. As such airmasses encounter the steep orographic rise along the Norwegian coast, they can yield

abundant precipitation (Stohl et al., 2008; Azad and Sorteberg, 2017). Past studies have emphasized the long-range transport characteristics, and their connection to the large-scale atmospheric flow configuration during such AR events. From a model study using artificial water tracers, Sodemann and Stohl (2013) estimated that 30-50 % of the precipitation from AR events could be from latitudes S of 40 °N. However, an observational confirmation of such model-derived estimates currently remains elusive.

The use of precipitation isotopes to gain information at the time scale of weather-systems dates back to the pioneering study of Dansgaard (1953), which suggested that the $^{18}$O-abundance in warm-frontal precipitation could be explained by a distinct fractionation process and below cloud evaporation. Since then, numerous studies have investigated the variation in precipitation isotopes between weather events and at different locations. Studies reveal that the isotope composition can vary substantially over short time scales. For example, analyses of single rainfall events have revealed variations in $\delta$D of between

7 ‰ for the case of southeast Australia (Barras and Simmonds, 2009) and 58 ‰ in California at sub-hourly time resolution (Coplen et al., 2008). A higher-resolution study in Cairns, Australia measured variations of up to 95 ‰ within a single 4-h period (Munksgaard et al., 2012). Several typical intra-event trends, such as "L", "V", and "W" shapes, have been identified by Muller et al. (2015). Despite numerous observations of the evolution of the isotope composition in rainfall over time and corresponding interpretation, it remains unclear how to separate the highly convoluted signal into the contribution from weather

system characteristics, moisture sources, and below-cloud effects.

The complexity of the isotopic information contained in rainfall at the event time scale has lead to a scientific controversy regarding the interpretation of the isotope signal during AR events. Coplen et al. (2008) (henceforth C08) sampled the precipitation during a land-falling AR at the coast of southern California at a time resolution of 30 min. C08 interpreted the isotope variation in rainfall during the event in relation to cloud height, using a Rayleigh distillation model. Coplen et al.





(2015) expanded the dataset and interpretation to numerous additional events. Investigating the same event as C08 with an isotope-enabled weather prediction model, Yoshimura et al. (2010) (henceforth Y10) instead emphasized the roles of horizontal advection and post-condensational processes for the temporal evolution of the precipitation isotope signal. Using the simultaneous water vapour and precipitation isotope measurements in this study, we attempt to shed new light on this so-far

unresolved controversy.

Here we present the analysis of highly resolved measurements of the stable isotope composition in precipitation and water vapour collected during a land-falling AR event in southwestern Norway during winter 2016. Thereby, we utilize a combination of observational and numerical methods, aiming to separate the moisture source information from effects related to moisture transport and precipitation processes.

In order to disentangle different factors that contribute to the isotope signal in precipitation, we adopt here a perspective where three sets of factors pertaining to the atmospheric water cycle can potentially have an influence. We hereby use the common $\delta$ notation as $\delta = \frac{R_{\text{sample}} - R_{\text{VSMOW}}}{R_{\text{VSMOW}}} \cdot 1000\ ‰$, where $R$ (e.g. $^2R = \frac{[\text{HD}^{16}\text{O}]}{[\text{H}_2^{16}\text{O}]}$) is the isotope ratio, to quantify enrichment or depletion with respect to the Vienna Standard Mean Ocean Water standard (VSMOW) (Mook and De Vries, 2001; IAEA, 2009).

(1) Depletion of heavy isotopes due to an atmospheric distillation or rain out process. The rainout history during the transport is essentially depending on the temperature difference between the moisture source and the condensation height above the precipitation site. This has been historically known as the rainout effect and can be described with a Rayleigh distillation model (Dansgaard, 1964). A larger temperature difference leads to a greater rain out process and thus a more depleted isotope profile in the condensate, that ultimately translates to the precipitation. For example, Dansgaard (1953) explained the gradual

enrichment of $^{18}$O-abundance in the precipitation from a warm front with decreasing of the condensation temperature as the front passes the observation site.

(2) Ocean-atmosphere conditions at the moisture source affect the isotope composition of generated water vapour (Gat, 1996). The deviation from equilibrium fractionation during evaporation at the source can be quantified by the $d$-excess parameter, calculated as $d\text{-excess} = \delta\text{D} - 8 \cdot \delta^{18}\text{O}$ (Dansgaard, 1964). Specifically, theoretical studies and observations have shown

that $d$-excess in the generated vapour over ocean surface is dependent on relative humidity (RH) with respect to sea surface temperature (SST), and to second-order to the SST itself in the source area (Merlivat and Jouzel, 1979; Uemura et al., 2008; Pfahl and Sodemann, 2014). As an example, high $d$-excess anomalies are usually observed in water vapour formed during so-called marine cold air outbreaks (Aemisegger and Sjolte, 2018; Aemisegger, 2018), where cold dry air moves over relative warm ocean waters and triggers strong evaporation (Papritz and Spengler, 2017; Papritz and Sodemann, 2018). In contrast, land

regions and more calm ocean evaporation are associated with lower $d$-excess (Aemisegger et al., 2014; Thurnherr et al., 2020). The $d$-excess is often assumed to be conserved during transport. However, microphysical processes within and below clouds can influence the $d$-excess in local precipitation, and thus obscure information on the evaporation conditions in the source area (Jouzel and Merlivat, 1984; Graf et al., 2019).

(3) Microphysical processes within clouds and post-condensational exchange processes of falling precipitation can alter

the isotope composition. While isotopic equilibrium can be assumed for rain formation in warm clouds, kinetic effect exists





at snow formation. Vapour deposition in a supersaturated environment with respect to ice, therefore, increases $d$-excess in precipitation (Jouzel and Merlivat, 1984). Liotta et al. (2006) proposed that higher $d$-excess also exists in orographic clouds since kinetic effects should be expected in the first step of droplet formation, while in-cloud droplets are short-lived, and thus can not reach equilibrium with the surrounding vapour. For deep convective systems, factors such as condensate lifting,

convective detrainment and evaporation in unsaturated downdrafts can play a critical role in the control of the isotope of precipitation (Bony et al., 2008). Below cloud processes have been noted in many precipitation events from previous studies (Dansgaard, 1953; Ehhalt et al., 1963; Miyake et al., 1968; Barras and Simmonds, 2009; Guan et al., 2013; Wang et al., 2016). Below-cloud evaporation usually dominates at the beginning of a precipitation event, when the atmosphere below cloud base is still unsaturated. As the atmosphere gradually gets saturated, the isotopic exchange between raindrops and surrounding vapour

intensifies (Graf et al., 2019). Depending on the intensity of below-cloud exchange processes, isotopes in precipitation can deviate more or less strongly from Rayleigh model expectations.

In the following, we present the unique dataset acquired during a land-falling frontal system, associated with an atmospheric river, at the end of the North Atlantic storm track. Using a combination of remote-sensing and in situ instrumentation (Section 2), we provide a detailed observation of meteorological parameters (Section 3) and the isotope composition in near-surface

water vapour and precipitation (Section 4) during a substantial precipitation event on 07 Dec 2016 at Bergen, southwestern Norway. We first quantify below-cloud exchange processes by means of the interpretative $\Delta\delta\Delta$D framework proposed recently by Graf et al. (2019) (Section 5.1). Then, we relate the observed evolution of the isotope signal to weather system characteristics (Section 5.2). We hypothesise that the remaining signal then reveals the source conditions in the $d$-excess parameter. We therefore compare our observational results with moisture source conditions and $d$-excess predictions obtained from a Lagrangian

moisture source diagnostic (Sodemann et al., 2008) and interpret the results in terms of assumptions and model deficiencies (Section 5.3). In a brief discussion, we attempt to contribute constructively to the dispute of C08 and Y10 (Section 6). Finally, conclusions are drawn in Section 7.

## 2 Data and methods

This section describes the measurement site, the installation and procedures used for acquisition of meteorological and isotopic

information, weather prediction model data, and of the method for diagnosing moisture sources.

### 2.1 Measurement site

Bergen is located at the coast of southwestern Norway (60.3837 °N, 5.3320 °E), with an annual mean temperature of 7.6 °C during 1961-1990 (*sharki.oslo.dnmi.no*). Being located at the end of the climatological North Atlantic storm track (Wernli and Schwierz, 2006; Aemisegger and Papritz, 2018), extratropical cyclones frequently bring moist airmasses to the Norwegian

coast. At the steep orographic rise from sea level to above 600 m in a distance of 2 km, the airmasses frequently produce intense precipitation. The average annual precipitation during 1961–1990 was 2250 mm, with the highest monthly average being 283 mm in September and lowest being 106 mm in May (Meteorologisk Institutt, *sharki.oslo.dnmi.no*).





Meteorological observations are performed operationally at the WMO station Bergen-Florida (ID 50540) at 12 m a.s.l. Additional measurements were acquired on the rooftop observatory (45 m a.s.l) of the Geophysical Institute (GFI), University of Bergen, located at a distance of 70 m from the WMO station. This additional instrumentation consisted of a Micro Rain Radar (MRR2, METEK GmbH, Elmshorn, Germany), a Total Precipitation Sensor (TPS-3100, Yankee Environmental Systems, Inc.,
USA), a Parsivel[2] disdrometer (OTT Hydromet GmbH, Kempten, Germany) and an automatic weather station (AWS-2700, Aanderaa Data Instruments AS, Bergen, Norway). A subset of these parameters (air temperature, pressure, RH, wind speed) from the AWS-2700 were consistent with the TPS-3100 and the WMO station measurements.

Precipitation rate was measured by 3 instruments. The TPS-3100 Total Precipitation Sensor is an automatic precipitation gauge that provides real-time solid and liquid precipitation rate at a 60 s time interval (Yankee Environmental Systems, Inc.,
2011). The laser-based optical distrometer Parsivel[2] provides the precipitation intensity at a 60 s time resolution, using measurements of particle size and particle fall speed (OTT Hydromet GmbH, 2015). Comparison of these high-resolution precipitation measurements located at the rooftop with the rain gauge measurement from the WMO station Bergen-Florida at ground level indicates that the TPS-3100 overestimates precipitation slightly (up to 10 %), while the Parsivel[2] clearly underestimates the precipitation intensity (up to 40 %; see Appendix A). Hereafter, we utilize the precipitation rates from the TPS-3100 for further
analysis.

In addition to precipitation rate, the Parsivel[2] distrometer provides drop size and velocity spectra by separating the precipitation into 32 size classes from 0.2 to 5 mm and 32 velocity classes from 0.2 to 20 m s$^{-1}$. The instrument has been configured to record raw spectra at a 60 s time interval. The raw number of particles are converted into a per-diameter-class volumetric drop concentration (mm$^{-1}$ m$^{-3}$), including corrections following Raupach and Berne (2015). The drop size distributions are
then characterized by the mass-weighted mean diameter $D_m$ (mm). The drop size distribution is an important precipitation characteristic, among others to evaluate the extent of below-cloud evaporation (Graf et al., 2019).

Continuous vertical profiling of the hydrometeors during the event was conducted using the vertical-pointing doppler radar MRR2. Previous studies have demonstrated the value of these observations for stable isotope analysis in precipitation (C08; Muller et al., 2015). Operating at 24 GHz, the radar measures the height-resolved fall velocity of the hydrometeors and other
derived parameters, such as height-resolved size distribution and liquid water content (METEK Meteorologische Messtechnik GmbH, 2012). Here, the MRR2 was set up with a vertical resolution of 100 m for its 32 range gates, resulting in a measurement range from 100 m to 3200 m. The high resolution in time and height enables monitoring of the phase and evolution of hydrometeors, and thus the evolution of melting layers (Battan, 1973; White et al., 2002, 2003).

## 2.2 Water vapour isotope measurements

The stable isotope composition of ambient water vapour was continuously measured with a cavity ring-down spectrometer (L2130-i, Picarro Inc., USA) from an inlet installed on the GFI rooftop observatory. Ambient air was continuously drawn through the 4 m long 1/4 inch unheated PTFE tubing with a flow rate of about 35 sccm. The inlet was shielded from precipitation with a downward-facing plastic cup.





The analyser was calibrated every 12 hours using a Standard Delivery Module (A0101, Picarro Inc., USA; hereafter SDM) and a high-precision vaporizer (A0211, Picarro Inc., USA). During the calibration, two laboratory standards bracketing the isotope composition of typical ambient vapour (GSM1: $\delta^{18}O = -33.07\pm0.02$ ‰, $\delta D = -262.95\pm0.30$ ‰; DI: $\delta^{18}O = -7.78\pm0.02$ ‰, $\delta D = -50.38\pm0.30$ ‰) were blended respectively with dry air supplied from a molecular sieve (MT-400-4, Agilent Inc., Santa Clara, USA). The generated standard vapour was then measured for 20 min each at a humidity level of $\sim$20 000 ppmv.

The vapour data are post-processed and calibrated according to the following steps. (1) The raw data are corrected for isotope ratio–mixing ratio dependency using the correction function in Weng et al. (2020), which was determined for the same analyser used here. (2) For each calendar month, SDM calibration periods are identified. Then, the median value of mixing ratio, $\delta^{18}O$ and $\delta D$ are obtained for each calibration period. The values that deviate from the median value by more than 0.5 ‰ in $\delta^{18}O$ or 4.0 ‰ in $\delta D$ are discarded to remove variations due to bursting bubbles and other instabilities. The remaining data for each period are then averaged and the standard deviation calculated. Calibrations were retained if at least 60 % of the calibration period were kept after quality control. (3) The vapour measurements were calibrated to SLAP2-VSMOW2 scale following IAEA recommendations (IAEA, 2009). To this end, the two nearest bounding calibrations of sufficient quality were identified for each calendar day and each standard. Finally, the calibrated vapour data are averaged at a 10-minute interval using centred averaging.

## 2.3 Precipitation isotope sampling and analysis

Liquid precipitation was sampled at the GFI rooftop observatory at high temporal resolution with a manual rainfall collector, similar to the setup used in Graf et al. (2019). The collector consists of a PE funnel of 10 cm diameter, which directs the collected water into a 20 mL open-top glass bottle. A total of 71 precipitation samples were collected during the 24-h sampling period between 00:00 UTC 07 December and 00:00 UTC 08 December 2016. The sampling interval was adjusted according to the precipitation intensity. Two samples were collected over a 105 min interval, 8 samples with 20–40 min intervals, and 61 samples with 10–20 min intervals (ref. supplementary material). The bottle and funnel were dried between each sample using a paper wipe. The sample was immediately transferred from the bottle to a 1.5 mL glass vial (part no. 548-0907, VWR, USA) and closed with an open-top screw cap with PTFE/rubber septum (part no. 548-0907, VWR, USA) to prevent evaporation until sample analysis.

The samples were stored at 4 °C before being analysed for their isotope composition at FARLAB, University of Bergen, Norway. During the analysis, an autosampler (A0325, Picarro Inc.) transferred ca. 2 µL per injection into a high-precision vaporizer (A0211, Picarro Inc., USA) heated to 110 °C. After blending with $N_2$ (Nitrogen 5.0, purity >99.999 %, Praxair Norge AS), the gas mixture was directed into the measurement cavity of a cavity ring-down spectrometer (L2140-i, Picarro Inc., USA) for about 7 min with a typical mixing ratio of 20 000 ppmv. To reduce memory effects between sample, two so-called wet flushes consisting of 5 min of vapour mixture at 50 000 ppmv were applied to the analyzer at the beginning of each new sample vial. Three standards (12 injections each, plus wet flush) were measured at the beginning and end of each batch, consisting typically of 20 samples (6 injections each, plus wet flush). The averages of the last 4 injections were used for





further processing. The measurement data were first corrected for isotope–humidity dependency using a linear correction for the analyzer obtained over a humidity range of 15 000–23 000 ppmv. Then, data were calibrated to the SLAP2-VSMOW2 scale following IAEA recommendations (IAEA, 2009) using two secondary laboratory standards (VATS: $\delta^{18}$O = $-16.47\pm0.02$ ‰, $\delta$D = $-127.88\pm0.30$ ‰; DI: $\delta^{18}$O = $-7.78\pm0.02$ ‰, $\delta$D = $-50.38\pm0.30$ ‰). The long term reproducibility of liquid sample

analysis at FARLAB has been estimated to 0.15 ‰ for $\delta^{18}$O and 0.66 ‰ for $\delta$D, resulting in a measurement uncertainty of 1.05 ‰ for $d$-excess.

## 2.4   The concept of equilibrium vapour

Due to equilibrium and kinetic isotopic fractionation during phase transitions, the isotope ratios in water vapour and precipitation can not be directly compared to one another. Instead, we use the concept of *equilibrium vapour* to compare the state

of both phases (e.g. Aemisegger et al., 2015). The equilibrium vapour from precipitation is the isotope composition of vapour that is in equilibrium with precipitation at ambient air temperature $T_a$. We calculate the equilibrium vapour of precipitation as

$$\frac{\delta_{\mathrm{p,eq}}}{1000} + 1 = \alpha_{\mathrm{l\to v}}(T_a)\frac{\delta_{\mathrm{p}}}{1000} + 1, \tag{1}$$

where $\alpha_{\mathrm{l\to v}}(T_a)$ is the temperature-dependent fractionation factor of the liquid to vapour phase transition following Majoube (1971). We quantify the difference between equilibrium vapour from precipitation samples and ambient vapour then as

$$\Delta\delta = \delta\mathrm{D_{p,eq}} - \delta\mathrm{D_v}, \tag{2}$$
$$\Delta d = d_{\mathrm{p,eq}} - d_{\mathrm{v}}. \tag{3}$$

While a similar notation can be defined for $\Delta\delta^{18}$O, we use the notation $\Delta\delta$ to refer to $\Delta\delta$D only. Using the above deviations from isotopic equilibrium, Graf et al. (2019) introduced a useful interpretative framework to quantify the effect of below-cloud processes on the isotope composition of ambient vapour and precipitation. This so-called $\Delta\delta\Delta d$-diagram quantifies

the deviation of $\delta$D and $d$-excess in the liquid from the vapour phase at ambient temperatures from isotopic equilibrium as indicators of evaporation and equilibration below cloud-base. We make use of this interpretative framework to quantify the below-cloud processes during the AR event studied here. In addition, we utilize a set of sensitivity studies with the Below-Cloud Interaction Model (BCIM, Graf et al., 2019) to identify the main influences during the case studied here in the $\Delta\delta\Delta d$-diagram. The sensitivity experiments are described in more detail in Appendix B.

## 2.5   Reanalysis and weather forecast data

The large-scale meteorological situation is depicted using the global ERA-Interim reanalysis data from the European Centre for Medium-Range Weather Forecast (ECMWF) re-gridded to a $0.75\times0.75°$ regular grid. Moisture transport is quantified by the integrated water vapour transport (IVT; e.g. Nayak et al., 2014; Lavers et al., 2014, 2016), whereas mean sea level pressure (SLP) is used to depicts the location of weather systems.

In addition, air temperature, solid and liquid precipitation, cloud water and cloud ice were extracted as profiles across all model levels from ERA5 reanalysis data (Hersbach et al., 2020) with a 1-h time resolution. Lastly, air temperature, horizontal





wind speed and relative humidity at different pressure levels, as well as surface precipitation were retrieved from the archive of operational Harmonie-Arome forecasts in the MetCoop domain (Bengtsson et al., 2017). Operational forecasts initialized during the period 06 to 07 Dec 2016 were retrieved from the publicly accessible archive (http://thredds.met.no).

Furthermore, Morphed Integrated Microwave Imagery for total precipitable water (MIMIC-TPW) available from the Co-
operative Institute for Meteorological Satellite Studies (CIMSS) are used to depict total column water at a given time instant. Satellite Meteosat Second Generation (MSG) imagery composited as Airmass RGB is used to link the actual airmass type and clouds to the modelled Meteorological situation. The colouring scheme for airmass interpretation was adapted from Zavodsky et al. (2013).

## 2.6   Lagrangian moisture source diagnostic

Moisture sources are a potential factor influencing the isotope composition in precipitation. Here we apply a quantitative Lagrangian moisture source diagnostic WaterSip (Sodemann et al., 2008) to diagnose the moisture sources for evaporation contributing to the AR event on 07 Dec 2016. The WaterSip method identifies moisture source regions and transport conditions from a sequentially weighted specific humidity budget along backward trajectories of air parcels that arrive over the target area.

More specifically, the method assumes that the change in specific humidity in an air parcel during each 6 h time step
exceeding a threshold value is due to either evapotranspiration or precipitation. A sequential moisture accounting then provides the fractional contribution of each evaporation event to the specific humidity at an air parcel location, and by taking into account the sequence of moisture uptakes and losses, the final precipitation in the target area. For the AR event in this study, the thresholds are set to be $0.2$ g kg$^{-1}$ for $\Delta q_{\mathrm{c}}$, with a 20-day backward trajectory length, and relative humidity >80 % to identify precipitation over the target region. These thresholds result in source attribution for over 98 %. Here, the moisture
uptakes from both within and above the boundary layer (BL) have been taken into account (Sodemann et al., 2008; Winschall et al., 2014).

The basis of the WaterSip diagnostic applied here is the dataset of Läderach and Sodemann (2016), which we have extended over the entire ERA-Interim period. In that dataset, the global atmosphere is represented by 5 million air parcels of equal mass calculated using the Lagrangian particle dispersion model FLEXPART V8.2 (Stohl et al., 2005), with wind and humidity and
other meteorological variables from the ERA-Interim reanalysis. For this study, the diagnostic was run with a target area of ca. $110 \times 110$ km centred over Bergen (59.9–60.9 °N and 4.3–6.3 °E), including both land and ocean regions. The precipitation event studied here was represented by in total 1100 trajectories arriving in the target area.

As with other methods to identify moisture source regions, the WaterSip diagnostic is associated with uncertainty due to threshold values, interpolation errors, and conceptual limitations (Sodemann et al., 2008; Sodemann, 2020). To enable a
comparison with stable isotope observations, the WaterSip method predicts the $d$-excess from the evaporation conditions at the moisture sources using the empirical relation of Pfahl and Sodemann (2014). More specifically, the SST over ocean regions and the surface specific humidity from ERA-Interim are used to calculate RH with respect to SST, and then to calculate $d$-excess from the empirical relation $d = 48.2 \, ‰ - 0.54 \, ‰/\% \cdot \mathrm{RH}_{SST}$, using a weighted average of all contributing moisture sources.





## 3 Meteorological situation

On 7 of December 2016 a substantial amount of precipitation accumulated over southwestern Norway. The precipitation was related to the influx of moist air from an AR, whose structure appears as a band of high vertically integrated water vapour (IWV) in passive microwave satellite imagery (Fig. 1a). The AR reaches as a narrow band from the central North Atlantic to the study region, impacting the entire west coast of southern Norway. At 12 UTC on 07 Dec 2016, the head of the AR has spread out broadly over the North Sea and the UK. The ERA-Interim reanalysis reproduces the observed structure of IWV faithfully, albeit with an apparent tendency to higher maximum values (Fig. 1b). While the IWV has commonly been used to define ARs, more relevant for the ensuing orographic precipitation is the associated water vapour flux, expressed as IVT (Lavers et al., 2014, 2016).

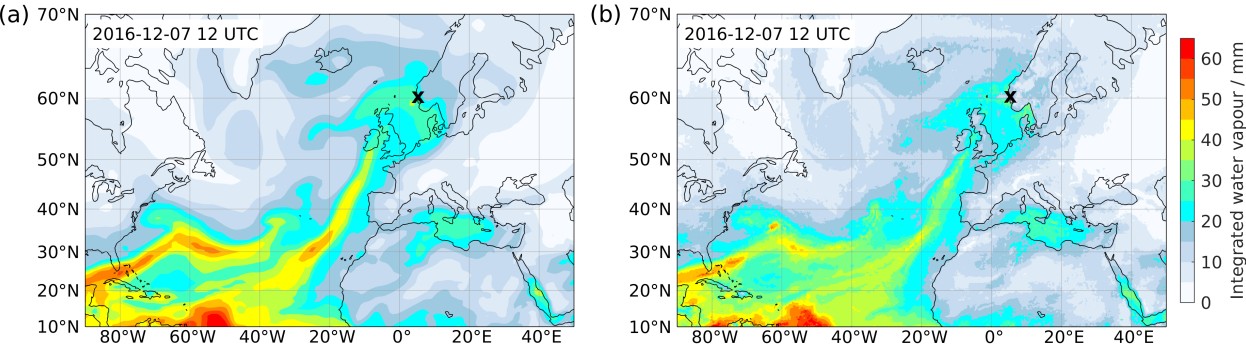

**Figure 1.** Vertically integrated water vapour (IWV) for the atmospheric river event occurring at 12 UTC 07 Dec 2016 in **(a)** Morphed Integrated Microwave Imagery at CIMSS (MIMIC-TPW) from the Cooperative Institute for Meteorological Satellite Studies (CIMSS) and **(b)** ERA-Interim analysis. The measurement site at Bergen is indicated by black cross.

The onshore flow of the large amounts of water vapour resulted in a prolonged precipitation event in Bergen, lasting from 00 UTC 07 Dec 2016 to 00 UTC 08 Dec 2016. Weather maps from the UK MetOffice show a sequence of surface warm fronts impinging upon southwestern Norway at 06 UTC on 07 Dec 2016 (Fig. 2a). This set of fronts is attached to a cyclone south of Iceland with core pressure of $985\,\mathrm{hPa}$. The fronts are embedded in a pronounced westerly flow, bounded by a broad anticyclone with a centre over southeastern Europe and a core pressure of $1039\,\mathrm{hPa}$. The individual warm fronts have approached one another over several days (not shown). We note that in the present case, the onshore water vapour flux is enhanced by the pressure gradient between the Icelandic low and the high-pressure over Europe. Similar configurations have been observed earlier to be associated with AR events in coastal western Norway (Azad and Sorteberg, 2017).

At 06 UTC on 07 Dec 2016, the first front has passed over land, as seen by the $850\,\mathrm{hPa}$ temperature north of Ålesund (Fig. 2c) and the widespread precipitation above $2\,\mathrm{mm\,h^{-1}}$ (Fig. 2d) obtained from the control forecast of the AROME MEPS regional forecasting system. The trailing warm front is still at a distance from the coastline, but already causes intense precipitation near



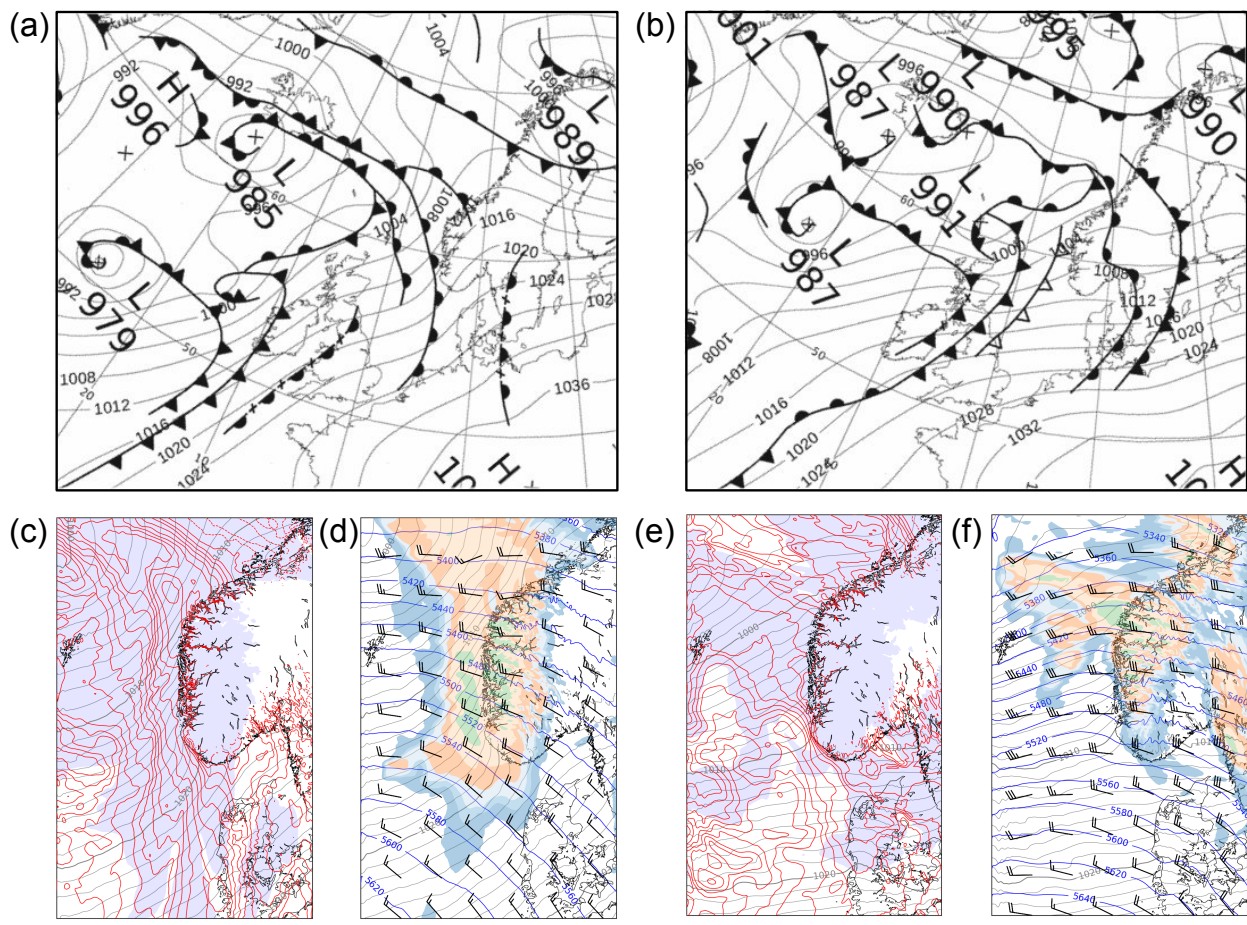

**Figure 2.** Overview of frontal structures during the precipitation event on 07 Dec 2016. Sea level pressure and surface fronts identified by the UK Met Office at **(a)** 06 UTC and **(b)** 18 UTC. **(c)** Sea level pressure (hPa, grey lines), air temperature (K, red lines), and relative humidity above 80 % (shaded) at 850 hPa at 06 UTC. **(d)** Sea level pressure (hPa, grey), 500 hPa geopotential height (g.p.m, blue), wind barbs at 500 hPa, and 1 h accumulated precipitation (mm, shaded) at 06 UTC. **(e)** As panel (c), but at 18 UTC. **(f)** As panel (d) but at 18 UTC. Panels (c) and (d) are from the 12 h MEPS forecast initialized at 18 UTC on 06 Dec 2016. Panels (e) and (f) are from the 06 h MEPS forecast initialized at 12 UTC on 07 Dec 2016.





the coast (Fig. 2d, green shading). At 18 UTC on 07 Dec 2016, the Icelandic cyclone has started to fill in, with the warm frontal system dissolving over southern Scandinavia. An upper-level cold front, trailed by a surface warm front approach the coast of southwestern Norway at this time (Fig. 2b). The temperature at 850 hPa shows the transition to a more cloud-free area with variable gradients as the upper-level cold air arrives over the North Sea (Fig. 2e). While there is still widespread precipitation

over southern Norway, a more scattered precipitation regime sets in at this time (Fig. 2f).

## 3.1 Meteorological surface observations

Meteorological surface observations from the tower observatory are displayed for the AR event, lasting from 00:00 UTC on 07 Dec to 00:00 UTC on 08 Dec 2016 (Fig. 3). The local pressure at the height of the observatory gradually dropped from 1015 hPa at the start of the event to 997 hPa at 00 UTC on 08 Dec (Fig. 3a, blue line). As the warm airmass approached, the air

temperature at the tower station gradually increased from 5.0 °C at 05:00 UTC on 07 Dec 2016 to 11.0 °C at 00:00 UTC 08 Dec 2016 (Fig. 3a, black line).

Precipitation already started forming before the increase of temperature, with precipitation rate from TPS-3100 (Fig. 3b, black line) below $1\,\mathrm{mm\,h^{-1}}$ between 00:00 and 03:30 UTC. Precipitation then steadily increased to $5.5\,\mathrm{mm\,h^{-1}}$ at 07:00 UTC, and varied thereafter on a generally high level throughout the rest of the day, with a brief intermission at 12:00 UTC, and

ending on 23:30 UTC. Rainfall became in particular more variable after 14:30 UTC, reaching brief maxima above $7.0\,\mathrm{mm\,h^{-1}}$. The total precipitation amount during this 24-h event was 55.3 mm. Other instruments for precipitation measurements provide a similar time series of precipitation intensity, and comparable precipitation totals (Appendix A).

Relative humidity changed markedly during the event. Before 04:30 UTC, RH varied between 77 and 80 %. As precipitation intensified, and before the temperature started to increase at 05:00 UTC, RH gradually increased to 92 % at 09:00 UTC, and

remained between 92 and 95 % thereafter (Fig. 3b, blue line).

According to the time evolution of the meteorological parameters presented above, in particular the radar reflectivity, we separate the AR event into 4 distinct precipitation stages: pre-frontal Stage I before 03:30 UTC (purple bar), first frontal Stage II between 03:30 and 07:00 UTC (blue bar), a second frontal Stage III between 07:00 and 14:30 UTC (red bar), dominated by stratiform precipitation processes, and a post-frontal Stage IV after 14:30 UTC (yellow bar) that is dominated by convective

precipitation. The four stages are indicated with corresponding colour bars at the top and bottom of Fig. 3.

The drop size distribution followed a similar evolution as the precipitation rate (Fig. 3c). At the beginning of the event, raindrop number concentration maxima were small, with the drop size maximum near 0.4 mm (Fig. 4a, Stage I). The drop size spectra started to show a more pronounced peak from 01:30 UTC, as well as an increase of raindrop number concentrations (Fig. 4a, Stage II). On some occasions during Stage II, a bi-modal distribution in drop sizes was observed. Drop size spectra

had pronounced maxima at the smallest drop size categories between 09:00 and 11:00 UTC, and became broader between 13:00 to 14:30 UTC (Fig. 4a, Stage III). A small number of large raindrops (>1 mm) had appeared during Stage II and III. The large raindrops had disappeared after entering Stage IV, except for some intense precipitation periods between 18:30 and 20:20 UTC, around 21:30 UTC, and around 22:40 UTC. A particular feature for Stage IV is that the amount of large raindrops



**Figure 3.** Time series of observations at ∼45 m a.s.l. in Bergen between 00 UTC 07 December and 00 UTC 08 December 2016. **(a)** Local temperature (black line) and air pressure (blue line) from the automatic weather station (AWS-2700). **(b)** 10 min averaged rain rate from the Total Precipitation Sensor (grey shading) and relative humidity from AWS-2700 (blue line). **(c)** Droplet number concentrations from the Parsivel². **(d)** Reflectivity from the Micro Rain Radar. **(e)** $\delta$D of the 10 min averaged vapour (grey dots) and $\delta$D of the equilibrium vapour from precipitation (black segments). The uncertainties are 0.60 ‰ and 0.11 ‰ for $\delta$D of vapour and of the equilibrium vapour from precipitation, respectively. **(f)** Same as in **(e)** but for $d$-excess, including $d$-excess of precipitation (blue segments). The uncertainty is 0.83 ‰ for $d$-excess of vapour, and 0.20 ‰ for $d$-excess of the equilibrium vapour from precipitation and precipitation. Precipitation periods I–IV are indicated with color bars at top and bottom of the figure.





(0.5–1.0 mm) increase substantially at the expense of raindrops with $<0.5\,\mathrm{mm}$ diameter (Fig. 4a, Stage IV). This feature is likely to be associated with the shift from stratiform to convective precipitation.

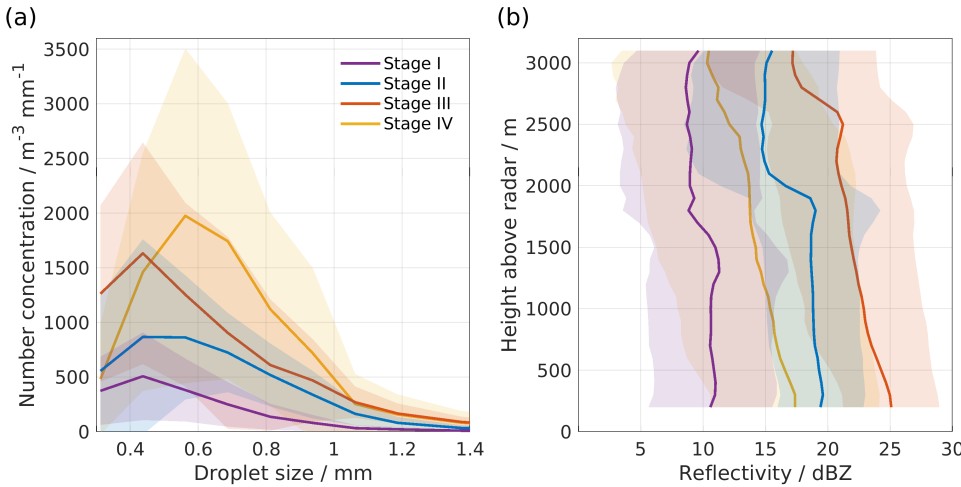

**Figure 4.** Averaged **(a)** number concentration of rain droplet per droplet size, and **(b)** reflectivity profile from the Micro Rain Radar at each precipitation stage during the AR precipitation event on 07 December 2016. The shading indicates one standard deviation. The lowermost layer of the reflectivity profiles has been removed due to ground clutter.

The vertical pointing MRR2 reveals hydrometeor profiles and melting layer height during the event (Fig. 3d). Before 03:30 UTC, precipitation was weak and did not continuously reach the surface, indicating the presence of evaporation of falling hydrometeors, or below-cloud evaporation. The overall reflectively is low at this stage (Fig. 4b, Stage I). As precipitation gradually intensified after 03:30 UTC, a melting layer started to appear, as well as ice-phase hydrometeors aloft, in particular after 05:00 UTC. The melting height increased from 1600 to about 1900 m between 03:30 and 04:30 UTC and stayed there until 07:00 UTC. The melting layer height then increased substantially to 2500 m at 07:00 UTC, thereafter varying between 2500 and 2700 m until 14:30 UTC, with two precipitation gaps at 11:00 and 12:00 UTC. The increase of the melting height between Stage II and III is also clearly reflected on the averaged MRR2 profiles (Fig. 4b, Stage II and III). At 07:00 UTC, the second warm front arrives over the measurement location, in close agreement with surface frontal charts and regional weather prediction model forecasts (Fig. 2). Notably, the transition to the second warm front is almost undetectable in surface temperature, precipitation and relative humidity. It is also worth to note that during the periods of most intense precipitation (i.e. between 06:30 and 11:20 UTC, and between 13:30 and 14:30 UTC), an increase in reflectivity below 500–1000 m indicates droplet growth in the lowest 1500 m above ground (Fig. 4b, Stage III), underlining the importance of water vapour in lower atmospheric layers for the surface precipitation. Almost instantly after 14:30 UTC and until the end of the event, precipitation becomes more intermittent, and no more melting layer was detected (Fig. 4b, Stage IV). This swift change reflected the shift from a more stratiform phase, dominated by warm frontal airmasses, to a dominantly convective phase of the precipitation event as the upper-level cold front arrives (Fig. 2c, e). We speculate that the melting layer vanishes either because the convection was





too shallow to reach above the 0 °C isothermal line, or because the precipitation was too intermittent to expose a clear melting layer.

## 4 Observed stable isotope signature in vapour and precipitation

The measured isotope composition in the surface vapour and precipitation samples is now compared in relation to the four precipitation stages identified above. For the surface vapour, the $10\,\mathrm{min}$ averaged $\delta D_v$ initially showed a relatively stable value of $-120\,‰$ at Stage I (dotted line, Fig. 3e). Then $\delta D_v$ gradually decreased at the start of Stage II (03:30 UTC), until reaching a minimum of $-185\,‰$ at the end of this stage (07:00 UTC). At Stage III, corresponding to the arrival of the second, merged warm front, the value gradually returns to a less depleted level of $-160\,‰$ at 09:00 UTC and then varies between $-160$ and $-145\,‰$ until 13:30 UTC. As the upper-level cold front arrives, the precipitation regime is about to change from stratiform to a more convective regime, at which the $\delta D_v$ first drops to a secondary minimum of $-172\,‰$, before during Stage IV (after 14:30 UTC) increasing again first rapidly, then more slowly to $-110\,‰$ around 18:00 UTC and finally $-100\,‰$ after 21:00 UTC (the least depleted values of the event). The resulting stretched-out "W" shape of the water vapour isotopes resembles earlier observations made in precipitation samples (Muller et al., 2015). The amplitude of $72\,‰$ is substantial but smaller than for example observed in rainfall by C08. The relative evolution of $\delta^{18}O_v$ closely follows that of $\delta D_v$ (not shown).

The equilibrium vapour from precipitation $\delta D_{p,eq}$ approximately follow the pattern of surface vapour (Fig. 3e, black segments). There surface vapour isotope signal appears to lag the isotope signal in precipitation by about 30 min. Comparison of specific humidity from the isotope spectrometer with specific humidity calculated from the AWS shows no apparent time lag or offset at 1-min measuring frequency, indicating that atmospheric effects cause this time lag. Overall, the $\delta D_{p,eq}$ is more variable than the $\delta D_v$ time series. At Stage I, the isotope signal in $\delta D_{p,eq}$ is substantially less depleted than $\delta D_v$. This reverses at the beginning Stage II (after 03:30 UTC), and during the transition to Stage III, $\delta D_{p,eq}$ reaches a minimum, before it again is less depleted than $\delta D_v$ until about 08:30 UTC. Thereafter, differences between $\delta D_v$ and $\delta D_{p,eq}$ are small. An exception is the last hour of Stage III from 13:30 to 14:30 UTC, where $\delta D_{p,eq}$ is highly variable, and more depleted than $\delta D_v$. Right at the beginning of Stage IV, the $\delta D_{p,eq}$ is again more enriched than $\delta D_v$, before approaching equilibrium after about 18:00 UTC. The time offset, and the relative enrichment and depletion characteristics of vapour and precipitation are further examined in Sect. 5.

We now investigate the time evolution of the secondary isotope parameter $d$-excess in vapour and precipitation. After a value of $11\,‰$ during Stage I, the surface vapour $d$-excess ($d_v$) increases to $14\,‰$ at Stage II, and stays around that level until the beginning of Stage III at 08:00 UTC, one hour after the second warm front arrives (Fig. 3f, dotted line). Then the $d_v$ gradually decreases throughout the rest of the event, with a more rapid decrease from about $10\,‰$ as the upper-level cold front arrives at 14:30 UTC, to $d_v$ varying around $4\,‰$ between 18:00 and 21:30 UTC and eventually reaching $0\,‰$ at 23:00 UTC.

The $d$-excess of the equilibrium vapour from precipitation $d_{p,eq}$ shows a remarkable difference to $d_v$ at the beginning of the event (Fig. 3f, Stage I and II, black line segments). Here, $d_{p,eq}$ are substantially lower than the $d_v$, with the lowest values even being negative ($-7$ and $-9\,‰$) during Stage I. This results in a large difference between $d_v$ and $d_{p,eq}$ of 18 and $20\,‰$,





respectively. During Stage II, $d_{\mathrm{p,eq}}$ gradually approaches $d_{\mathrm{v}}$, remaining about 2–4 ‰ lower than $d_{\mathrm{v}}$. Similar to $d_{\mathrm{v}}$, $d_{\mathrm{p,eq}}$ then shows a continuous decrease between 07:00 UTC and 16:30 UTC, then stabilising (with some variability) around 2 ‰. The original $d$-excess of precipitation, $d_p$ (Fig. 3f, blue line segments), should theoretically be equal to $d_{\mathrm{p,eq}}$. Small discrepancies at Stage I, Stage IV, and the two depletion minima, may at least partly arise from the definition of the $d$-excess (Dütsch et al.,

5 2017).

As is evident from the results presented above, the precipitation and vapour isotope measurements, especially when combining $\delta$D and $d$-excess parameters, clearly provide signals that are not apparent in standard meteorological observations (such as air temperature and precipitation rate). Following our hypothesis that the isotope signature at each stage reflects the impact of several atmospheric processes, including moisture origin, processes during advection and mixing, condensation processes

in clouds, as well as below cloud interaction, we now attempt to disentangle the individual contributions from these processes on the observed isotope signature at the surface during the AR event.

## 5   Impacts on the stable water isotope signature

The precipitation isotope signal during a weather event results from a convolution of different processes. We now proceed backwards from the last process, the below-cloud interaction, to weather system and transport influences, to the moisture

source signal, to investigate how different processes contribute throughout the event.

### 5.1   Contribution from below-cloud interaction processes

Below-cloud interaction processes consist of the continuous exchange of falling precipitation below cloud base with the surrounding vapour in the atmospheric column. In near-saturated conditions, liquid precipitation will exchange with surrounding vapour in a near-equilibrium process. In undersaturated conditions, the vapour exchange will lead to a net mass loss of the

droplets. Resulting from the same underlying process, both exchanges are strongly influenced by drop size, whereby smaller droplets being affected more strongly (Graf, 2017).

We investigate the change in isotope composition due to below cloud processes using the $\Delta\delta\Delta d$-diagram (Graf et al., 2019). The $\Delta\delta\Delta d$-diagram uses the differences between equilibrium vapour from precipitation and ambient vapour in terms of both $\delta$D and $d$-excess ($\Delta\delta$ and $\Delta d$, Sect. 2.3) as its axes (Fig. 5). The diagram is divided by the zero reference lines into

four quadrants. The closer data points are located near the origin, the closer the equilibrium between the vapour and liquid precipitation. Data points located in the lower right quadrant have positive $\Delta\delta$ and negative $\Delta d$ values, reflecting the impact of strong evaporation below cloud base. Conversely, data points in the lower-left quadrant have undergone moderate below-cloud evaporation and equilibration. Negative $\Delta\delta$ values indicate that below-cloud evaporation has been incomplete, and does not yet entirely overprint the more depleted isotope signal in the liquid precipitation at cloud base.

The temporal evolution of the precipitation samples during the AR event proceeds from the lower right quadrant, with the first to samples from Stage I displaying the strongest influence of below-cloud evaporation (Fig. 5a, letter A). Samples from Stage II are in the bottom left quadrant, first reflecting moderate below-cloud evaporation and some equilibration (letter B).





**Figure 5.** $\Delta\delta\Delta d$-diagram for precipitation samples collected during the AR event on 07 December 2016. Samples coloured according to **(a)** sampling start time (UTC), **(b)** relative humidity at the surface (RH$_s$, %), **(c)** precipitation rate ($RR$, mm h$^{-1}$), and **(d)** droplet mean diameter ($D_m$, mm). Letters in panel (a) mark time periods (see text for details). Grey lines in panels (b–d) show sensitivity experiments with the idealized below-cloud interaction model of Graf et al. (2019) regarding the parameters surface air temperature ($T_a$), cloud base height ($z_c$), and relative humidity at the surface (RH$_s$) with regard to a reference simulation where $T_a = 5$ °C, $z_c = 1500$ m, and RH$_s = 90$ % (see text for details).





Towards Stage III (08:30 UTC), samples are close to equilibrium with surface vapour, with slightly negative $\Delta d$ values (0 to $-4$ ‰) and a relatively large spread of both positive and negative $\Delta\delta$D values (12 to $-12$ ‰, letter C). An interesting phenomenon then occurs at the transition to Stage IV, when first a stronger cloud influence is apparent, with data points near $-10$ ‰ for $\Delta\delta$ (Fig. 5a, letter D), before directly jumping to $+10$ ‰ after 15:00 UTC (Fig. 5a, letter E). For the remainder of Stage IV, data

points then progressively move closer to equilibrium conditions, corresponding to the origin of the coordinate axes (letter F). Note that the samples from different stages are well separated in the diagram, indicating different dominating processes at each stage.

A key factor of influence for the below-cloud evaporation is RH below cloud base. When coloured by RH from the AWS, it is evident that the samples most affected by below-cloud evaporation coincide with below 90 % RH at the surface (Fig. 5b). At

90–95 % RH, the precipitation samples remain at non-equilibrium and reach the origin only for above 95 % RH. A sensitivity study with idealized simulations using BCIM (below-cloud interaction model, Graf et al., 2019) provides the coordinate system of drop-size dependent effects of RH on raindrops falling from 1500 m to the surface with initial conditions approximately resembling the situation during Stage I and II (see details in Appendix B). Albeit offset by about 10–15 % from observed RH, the sensitivity study shows a clear tendency towards lower $\Delta d$ with lower below-cloud RH.

While RH is a key driver of below-cloud interaction, several other factors are also important, for example, precipitation rate. The two samples with the lowest rain rates of about 0.5 mm h$^{-1}$ (during Stage I) are located in the lower right quadrant of the $\Delta\delta\Delta d$-diagram (Fig. 5c). Several subsequent samples with slightly higher rain rate ($\sim$ 0.9–2.2 mm h$^{-1}$) are located in the left quadrant, ranging from about $-15$ to $-6$ ‰ in $\Delta d$. As the rain rate of the sample further increases and the ambient air nearly saturates, the effect from below cloud evaporation weakens. Samples with relatively heavy rain rates (mostly between

3 and 5 mm h$^{-1}$) are found during the rest period of the event; they are located close to the zero $\Delta d$ line, indicating weak influences from below cloud interactions. A sensitivity analysis of the formation height parameter in the BCIM model shows weak sensitivity, that aligns horizontally along the $\Delta\delta$ axis with increasing height. Interestingly, this agrees with data points at the transition to Stage III when the melting layer was among the highest (Fig. 3d).

The small precipitation rates are also a consequence of the below-cloud evaporation in an undersaturated environment. This

below-cloud evaporation also leads to a reduced size of precipitation droplets, characterised by the droplet mean diameter. In the $\Delta\delta\Delta d$-diagram, the samples with the lowest rain rates also have a small droplet mean diameter of below 0.9 mm (Fig. 5d). There are further samples with mean diameters below 1 mm during Stage IV of the precipitation event. At these times, rather than being due to evaporation effects, the small drop sizes and the near-saturation conditions indicate that droplet growth may be taking place actively. An analysis of the sensitivity to the temperature profile with the BCIM shows a sloping of the

sensitivity from a horizontal to a diagonal orientation with warmer temperatures. This is in qualitative agreement with the observations during the event with surface warming continuing from Stage III through Stage IV.

In summary, we observe strong below cloud interaction at the beginning of the rainfall event. The period (Stage I and II) is characterised with the least saturated ambient air, the lowest rain rate, the smallest droplet size, and the lowest melting layer height. All these features except the melting layer height favour the occurrence of the below-cloud interaction. Transition phases

between stages increase the disequilibrium between surface vapour and precipitation, with the precipitation signal leading the





vapour in characteristic ways (Fig. 5a, letters A–F). The non-equilibrium fractionation during the evaporation causes the rain droplets to be more enriched in heavy isotopes (i.e., higher $\delta^{18}$O and $\delta$D values). At the same time, more HDO is transferred to the vapour phase, yielding to a low or even negative $d$-excess in the remaining rain droplet. These isotopic signatures match the precipitation samples taken during this period (Fig. 3e, f; Fig. 5). The variation during Stage III and IV, however, shows

that these two stages are less affected by below-cloud interactions, and more related to a change in parameters related to the weather system, such as formation height and the temperature profiles. We, therefore, focus now on the potential contribution of weather-system related changes to the isotope composition of surface vapour and precipitation during the AR event.

## 5.2    Weather-system contribution

Now, we use the 4 stages, defined based on the surface meteorological observations (Fig. 3) to investigate the relationship

between the observed isotope signatures and weather-system characteristics. The precipitation event was dominated by two warm fronts, passing over Bergen in close sequence (Sect. 3). The fronts are apparent as marked gradients in air temperature at 850 hPa around 06 UTC (Fig. 2c).

A more continuous display of the frontal passage is provided by a time-height cross-section of equivalent potential temperature ($\theta_e$), cloud water, and precipitation, using hourly ERA5 reanalysis data (Fig. 6). The cross-section depicts a constantly

increasing temperature on the surface (below 850 hPa), consistent with the surface meteorological observations (shading), as well as a descending cloud base (black dotted line). A relatively deep layer of cold air near the surface present at the beginning of Stage I is replaced by warmer and more humid air. The cloud base is initially near 850 hPa, as seen by the gradient in cloud water, just below the melting layer, which is at about 830 hPa at this time (purple solid line). Towards Stage II, there is an increasing contribution of ice-phase processes to the surface precipitation, with cloud ice of above $0.15\,\mathrm{g\,kg^{-1}}$ near 450 hPa

(white dotted lines). Snowfall rates increase from 0.1 to above $0.4\,\mathrm{mm\,h^{-1}}$ above the melting layer (white solid line), indicating riming of the ice particles between 600–750 hPa as an important contribution to the precipitation. The adequacy of this overall sequence is supported by the MRR2 radar observations (Fig. 3d) but indicates a delay of about 2–3 h in the ERA5 dataset.

The surface vapour and precipitation isotope composition during Stage I and II are initially dominated by below-cloud interaction. Both surface vapour and equilibrium vapour from precipitation exhibited relatively enriched $\delta$D (Fig. 3e), although

probably for different reasons. The low depletion of $-120\,‰$ in $\delta$D for the surface vapour is probably related to the pre-frontal boundary-layer airmass that originated from local evaporation, and had not undergone rain-out processes (e.g. a short distance moisture source). As identified in Sect. 5.1, the observed enrichment in the precipitation is probably the result of below-cloud evaporation, as reflected in the observed negative $d$-excess of $-6$ to $-9\,‰$ in the precipitation samples. With the precipitation signal leading the vapour isotope composition, the weather-system signal progressively becomes more dominant throughout

Stage II, levelling at $-180\,‰$ between 05:00 and 06:00 UTC. We consider this the actual $\delta$D isotope signature of the first frontal airmass.

From Stage I to Stage II, the $d$-excess of surface vapour increased from 12 to $15\,‰$. We consider two possible influencing factors for this increase. First, the increase could reflect the gradual shift from the pre-frontal to the newly arriving warm-frontal airmass. However, there was a large distance between the $d$-excess of equilibrium vapour from precipitation and that of



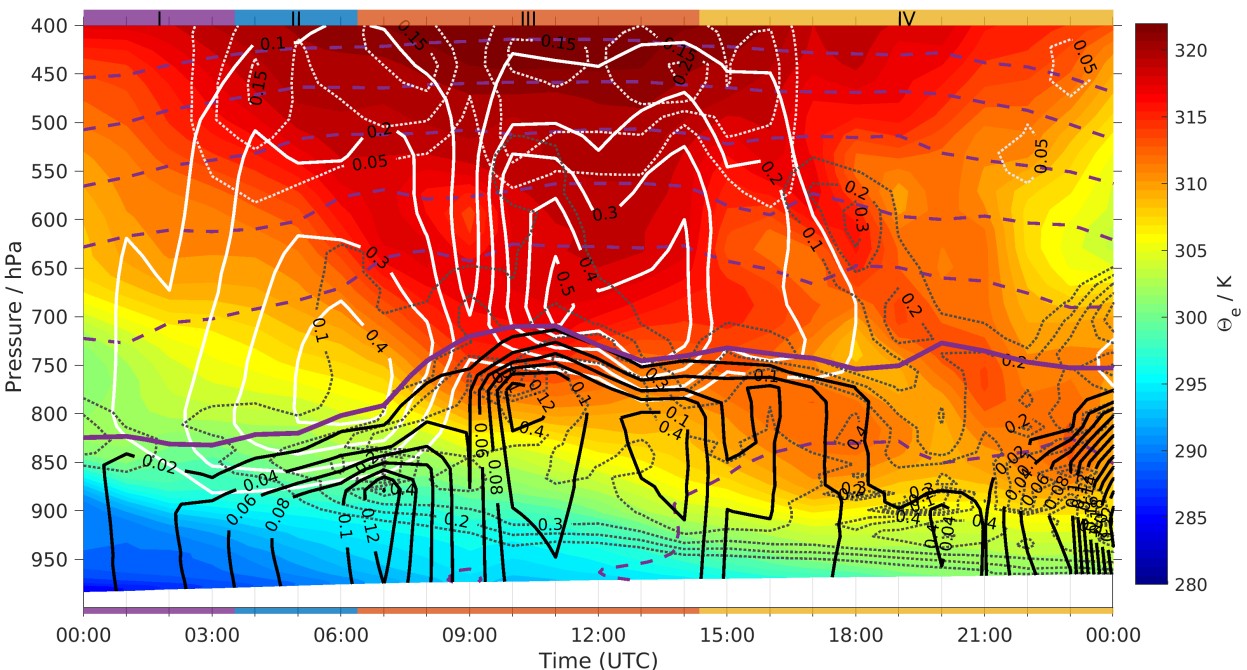

**Figure 6.** Hourly equivalent potential temperature from ERA5 reanalysis for the observation site at Bergen between 00 UTC 07 December and 00 UTC 08 December 2016. Solid white line indicates specific snow water content and dotted white line specific cloud ice water content. Solid black line indicates specific rain water content and dotted black line specific cloud liquid water content. The unit of all contours for different water species is $\mathrm{g\,kg^{-1}}$. Thick purple line indicates the $0\,^{\circ}\mathrm{C}$ isothermal line and dashed purple lines indicate isothermal lines deviating from $0\,^{\circ}\mathrm{C}$ isothermal line with $5\,^{\circ}\mathrm{C}$ intervals. Colour bars at top and bottom indicate precipitation periods I–IV.

surface vapour (up to $\sim$12 ‰). This indicates the influence of the below-cloud exchange. At the end of Stage II, the evolution of $d$-excess of equilibrium vapour from precipitation and $d$-excess of surface vapour converge, indicating a balance between column vapour and precipitation. Following this interpretation, we consider the $\sim$14 ‰ as the most likely value for $d$-excess signal of the first warm front.

5     The transition to Stage III with the second warm front is indicated by a substantial jump in melting layer height around 07–08 UTC to $700\,\mathrm{hPa}$ (Fig. 6, purple line), and a gap in snowfall and intensified precipitation around 09 UTC. At this time, the cloud becomes markedly deeper, and regions of cloud liquid and cloud ice overlap at $550\,\mathrm{hPa}$. Precipitation shows a maximum above $800\,\mathrm{hPa}$, and decreases below. This rain evaporation may be overestimated by the reanalysis since the precipitation radar instead shows an increase in reflectivity in the lowest $1000\,\mathrm{m}$ above the surface (Fig. 3d).

10     The isotopic signal of this second warm front is less depleted, and produces a transition to about $-160$ ‰ for $\delta\mathrm{D}$, led by the precipitation (Fig. 3e). In addition to being warmer, cloud processes extend over a deeper section of the lower and middle troposphere during the second front. The enriching trend probably corresponds to a gradual lowering of the effective





condensation level. The lowering here appears connected to the lower of cloud base height, allowing an increased contribution to falling raindrops that gain mass from, for example, the collision with droplets formed at low levels. Indeed, we observe a noticeable increase of radar reflectivity at the surface level below 1 km during Stage III (Fig. 3d and 4b). The contribution of low-level vapour to surface precipitation is also consistent with the arguments by Yoshimura et al. (2010) based on a regional

model study of an AR event that the precipitation isotope signal can be influenced by a deep section of the atmosphere.

The plateau in $\delta$D reached after about 09:00 UTC indicates that this likely is the actual isotopic signal of the second warm front. While both warmer temperatures and more contribution from lower atmospheric layers are consistent with the lower depletion, it is also possible that a different transport process has contributed to the different isotope signal of this airmass (see next section). The $d$-excess of both surface vapour and equilibrium vapour from precipitation during Stage III gradually

decreased from 15 to 9 ‰ for the vapour, and from 13 to 6 ‰ for precipitation. A plateau reached in the precipitation $d$-excess after 11:00 UTC indicates that the steady state in below cloud exchange has been reached thus the signal of the airmass is likely apparent at the surface level at this time.

In the ERA5 reanalysis, the middle and lower troposphere starts to become more unstable after 14:00 UTC, as indicated by $\theta_e$ changing from about 320 K to about 305 K towards the end of the day. Noting the shift by 3 h in relation to observations,

the transition to Stage IV is marked by the disappearance of ice-phase precipitation, with a tongue of cloud water reaching above 600 hPa, and cold air overrunning the warm front at about 720 hPa at 18:00 UTC (Fig. 2b). The such created instability may explain the very intense precipitation lasting for a 1-h period at the end of Stage III, associated with strong deviations in the $\Delta\delta\Delta d$-diagram. The local $\delta$D minimum of $-175$ ‰ at the transition of Stage III to Stage IV would then represent a higher-elevation cloud signal, reflecting the isotopic gradients in the column.

The stable stratification weakens further during the remainder of Stage IV, leading to a change from stratiform to convective precipitation. Precipitation formation shifts to the lower troposphere, mostly below the melting layer height, consistent with MRR2 measurements (Fig. 3d). The apparent lack of a melting layer implies condensation temperatures above 0 °C. The $\delta$D of both surface vapour and equilibrium vapour from precipitation gradually becomes less depleted, reaching $-110$ ‰ around 18:00 UTC and finally $-100$ ‰ after 21:00 UTC, even less depleted comparing with the values during Stage I (Fig. 3e). The

increased $\delta$D values reflected the shift to precipitation formation dominated by low-level water vapour. The $d$-excess plateaus at about 4 ‰ after about 16:00 UTC, with the equilibrium vapour trending towards 0 ‰ towards the end of the event. With the cloud water isolines nearing the surface, and near-saturated conditions in observations, the isotope signal essentially reflects conditions within a condensing airmass.

To understand the isotope signals of surface precipitation with a rain out perspective, we modelled the observed $\delta$D of surface

precipitation at different stages using the Rayleigh fractionation model of Jouzel and Merlivat (1984). The model assumes that ice crystals are formed below 0 °C under the condition of supersaturation over ice, thus including a kinetic effect during vapour-ice phase transition. Supersaturation is represented with a linear formula $S_i = 1 - 0.004T$ (T in °C) after Risi et al. (2010). The initial conditions are taken from global average conditions according to Craig and Gordon (1965) as $T_0 = 20$ °C, $\mathrm{RH}_0 = 0.75$, $\delta^{18}O_0 = -13$ ‰, $\delta D_0 = -94$ ‰. Precipitation is then produced by atmospheric vapour condensation under (pseudo)-adiabatic



conditions during airmass ascent. The condensation temperature of the precipitation is obtained when the modelled $\delta$D became equivalent to the observed $\delta$D in surface precipitation. The model results are shown in Table 1.

**Table 1.** The observed precipitation rate (RR) and isotope compositions ($\delta$D, $d$-excess), and the corresponding model estimate of condensation temperature ($T_c$), condensation height ($Z_c$), and $d$-excess ($d_c$) of the surface precipitation during the AR precipitation event on 07 Dec 2016 in Bergen. The model estimates are calculated using the observed $\delta$D values of the surface precipitation, according to a Rayleigh fractionation model of Jouzel and Merlivat (1984). Supersaturation over ice $S_i$ is assumed to occur during ice formation and is represented with a linear formula $S_i = 1 - 0.004T$ (T in $^\circ$C) after Risi et al. (2010). Input conditions have thereby been taken from global average conditions according to Craig and Gordon (1965) as $T_0 = 20\,^\circ$C, $RH_0 = 0.75$, $\delta^{18}O_0 = -13\,‰$, $\delta D_0 = -94\,‰$.

| | From (UTC) | To (UTC) | RR (mm) | $\delta$D (‰) | $d$ (‰) | $T_c$ ($^\circ$C) | $Z_c$ (m) | $d_c$ (‰) |
|---|---|---|---|---|---|---|---|---|
| **Stage I** | 00:00 | 03:30 | 1.8 | $-14.9$ | $-3.2$ | 14.1 | 1280 | 11.7 |
| **Stage II** | 03:30 | 06:00 | 3.4 | $-76.0$ | 4.4 | 0.9 | 3900 | 9.9 |
| **1st minimum** | 06:00 | 06:50 | 2.0 | $-101.2$ | 8.2 | $-4.1$ | 4790 | 1.6 |
| **Stage III** | 08:30 | 13:15 | 13.8 | $-68.3$ | 8.3 | 2.4 | 3600 | 10.1 |
| **2nd minimum** | 13:35 | 14:15 | 2.4 | $-85.7$ | 3.8 | $-2.3$ | 4480 | 0.2 |
| **Stage IV** | 17:00 | 21:45 | 17.0 | $-16.7$ | 5.0 | 13.6 | 1380 | 11.7 |
| **Entire event** | 00:00 | 21:45 | 55.3 | $-51.9$ | 6.2 | 5.8 | 2970 | 10.7 |

The modelled condensation temperature of Stage I reaches above 14 $^\circ$C, substantially higher than the actual surface temperature ($\sim$5 $^\circ$C). With a condensation temperature well below $\sim$5 $^\circ$C, the $\delta$D of formed precipitation is expected to be quite

depleted. The modelled $d$-excess from the Rayleigh model is 11.7 ‰, in large contrast to the observed $-3.2$ ‰. The observed enriched $\delta$D and negative $d$-excess indicates that the cloud signal of precipitation has been substantially modified by below cloud evaporation. At Stage II, the modelled condensation temperature dropped to 0.9 $^\circ$C, which is in better agreement with the concurrent temperature profile (Fig. 6). The reduced difference between modelled and observed $d$-excess supports a lower influence from below cloud evaporation. However, cloud tops in ERA5 reach temperatures below $-25\,^\circ$C, which is not reflected

in the Rayleigh model. In Stage III, the modelled condensation temperature increases to 2.4 $^\circ$C, corresponding to the more enriched $\delta$D in precipitation. The modelled $d$-excess of 10.1 ‰ agrees well with the observed value of 8.3 ‰. Expecting very limited below cloud evaporation during Stage III, the overall good agreement between model and observation may indicate that the moisture origin for the water vapour at this stage is to a good extent represented by the model initial conditions (i.e. an ocean surface vapour mass with $T_0 = 20\,^\circ$C, $RH_0 = 0.75$, $\delta^{18}O_0 = -13\,‰$, and $\delta D_0 = -94\,‰$). Still, the relatively warm

condensation temperature compared to the depth of the stratiform cloud underlines that lower atmospheric layers contribute substantially to the rainfall total. At Stage IV, the modelled condensation temperature is again far too warm, with 13.6 $^\circ$C. The overestimation probably reflects that more local evaporation conditions should be used in this case. The modelled condensation temperatures for the two most depleted $\delta$D periods are $-4.1$ and $-2.3\,^\circ$C respectively. The low $d$-excess from the model may be associated with the high sensitivity of $d$-excess on the representation of supersaturation conditions during ice formation



in cloud (Jouzel and Merlivat, 1984) and should be considered with caution. Also in these two most depleted situations, the condensation temperature from the Rayleigh model is more consistent with a mass-weighted average of condensation, rather than cloud-top temperatures.

It is also worth to note that the precipitation amount collected during Stage I and II only contributes about 9.4 % of the total precipitation amount collected during the entire event. Hence the effect of below cloud evaporation will unlikely be detected in a precipitation sample that is collected on an event basis, or daily and longer time scales.

Based on the isotope signals of the different airmasses during Stage II to IV, we now explore to what extent these reflect the moisture source and transport conditions.

## 5.3 Relation of moisture sources to meteorological evolution

We now consider the synoptic development over the three days proceeding the precipitation event, with a focus on how moisture sources and moisture transport to Bergen are connected to the weather system configuration.

On 4 December 2016, two low-pressure systems are located south of Greenland and in the North Atlantic. Strong moisture transport takes place at the southern flank in the warm sector region, displayed as IVT above $800 \, \text{kg} \, (\text{ms})^{-1}$ (Fig. 7a). This region is connected to widespread cloudiness at the northern edge of an airmass with high humidity (Fig. 7b, green shading). Bergen (red cross) is under the influence of a weak pressure gradient, with an onshore flow from NE, and lower humidity. Moisture uptakes contributing to precipitation in Bergen during the AR event are identified for the respective time periods. The most substantial moisture uptake (thick blue contours) contributing to the precipitation on 07 Dec 2016 coincides with the boundary between the dry and cold air to the north (Fig. 7b, red and blue shades), and the moist airmass to the south (green shades) over the central and western North Atlantic. At this boundary, extensive cloud formation occurs, ranging from deep clouds (white) to low-level stratus clouds (light green).

On 5 December, the two low pressure systems have merged, with a core low below 975 hPa near Iceland (Fig. 7c,d). IVT and cloudiness in the frontal band have intensified. South of Norway and central Europe, high pressure is starting to form, with a 1030 hPa core pressure. The moisture uptake has moved further north and overlaps now with the IVT maximum. This warm frontal band coincides with the two warm fronts passing southern Norway during the event (Fig. 2a).

On 6 December at 12 UTC high wispy cirrus clouds mark the surface warm front over Bergen (Fig. 7f). The cyclone had entirely separated from its frontal bands and started to fill in. High pressure over Europe increased to 1040 hPa, with the pressure gradient further accelerating the onshore flow, supporting an intense meridional IVT of above $800 \, \text{kg} \, (\text{ms})^{-1}$, which just straddled over Scotland. Moisture sources advanced substantially further to the northeast, with the IVT maximum and now concentrated south of the British Isles.

On 7 December, a small, secondary cyclone dominated the moisture flux in the north, while the southern part of the IVT structure remained supported by yet another low-pressure system downstream (Fig. 7g). Moisture uptakes are identified over the North Sea near Scotland, contributing to precipitation in Bergen later that day (blue contours). The area over Scotland corresponded to relatively cold air with broken clouds intruding at the rear side, over the UK, belonging to the cold frontal air during Stage IV (Fig. 7h).

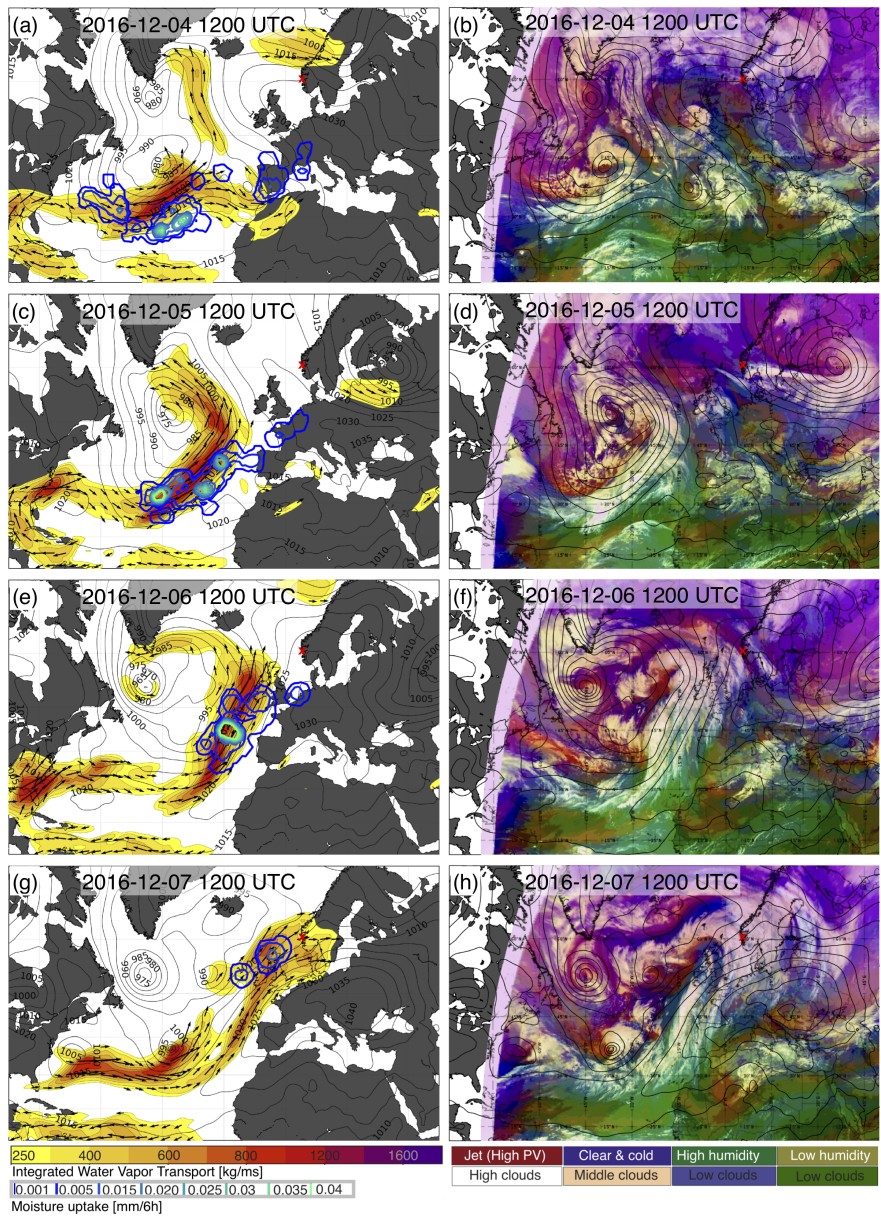

**Figure 7.** Synoptic situation at 12 UTC on the day of the precipitation event, 7th of December 2016 **(g,h)**, and three days prior **(a-f)**. Mean sea level pressure every 50 hPa in black contours. Left column: Integrated water vapour Transport (IVT, $\mathrm{kg\,(ms)^{-1}}$ >250 $\mathrm{kg\,(ms)^{-1}}$, filled contours), with the previously concurrent moisture uptake ($\mathrm{mm\,(6\,h)^{-1}}$, blue-green contours) that precipitated between 00 UTC on 7th December and 8th of December in the target area. Right column: Satellite Meteosat Second Generation (MSG) Airmass RGB (source http://eumetrain.org/). Colorbar adapted from airmass RGB interpretation EUMETSAT (Zavodsky et al., 2013). Red cross indicates the measurement site at Bergen.





In summary, moisture transport and moisture uptakes were clearly connected to the frontal structures during the AR event. The most substantial moisture uptake was occurring in the vicinity of the IVT maximum, embedded in the fused warm frontal bands. As the time window to the precipitation event shortened, the moisture uptake moved substantially further northward over the North Sea. This change in moisture source distance corresponds at least qualitatively to progressively less depleted

isotopic signature during the event. We now investigate more quantitatively how different the evaporation conditions at the moisture sources were for Stages II, III and IV.

### 5.3.1 Moisture source contribution

The evaporation conditions at the moisture sources identified above determine the vapour isotope composition before the start of the condensation processes. Here we investigate if the stepwise decrease in precipitation $d$-excess observed during Stage II

and Stage III can be related to changes in moisture source conditions. Moisture source conditions are quantified here in terms of moisture source distance, surface temperature, relative humidity with respect to sea surface temperature (Fig. 8a-c).

The large majority of moisture uptakes took place within a distance of 8000 km (Fig. 8a). The histogram for the main precipitation event at 12 UTC on 07 Dec 2016 is shown in grey shading, while the preceding time steps are shown in bold, and the later ones with dashed lines. During the sequence of the event, moisture sources shifted from local sources (less

than 1000 km distance during 00 UTC on 07 Dec 2016) to the most distant at 12 UTC, and finally again to closer locations (3000–4000 km distance), with a combination of local and remote sources at 00 UTC on 08 Dec 2016. An analysis of the corresponding moisture lifetime (not shown) provides the shortest lifetimes during the main precipitation phase at 12 UTC, with a median of about 3 days. This timing corresponding to uptake locations from 04 to 07 Dec 2016, shown in Fig. 7. In earlier and later stages, lifetime distributions also peak at less than 5 days, while including more notable contributions with

more than 5 days since evaporation.

Along with the shift in the moisture source location, evaporation conditions also changed. The most frequent temperature at the moisture sources was about 23 °C throughout the event, yet including a range of colder temperature conditions (Fig. 8b). Colder temperatures contributed in particular during the beginning of the event, when the average moisture source temperature was 17.6 °C at 00 UTC on 07 Dec 2016 (green line), and moisture sources were more local. Overall, the range of moisture

source temperature variations was relatively limited throughout the event (within 2 °C).

The relative humidity with respect to the SST ($RH_{SST}$) is a key factor in kinetic fractionation during evaporation (Craig and Gordon, 1965). Throughout the event, mean $RH_{SST}$ is around 65–70 % (Fig. 8c). The peak at near 100 % is an artefact of the contribution from land regions where $RH_{SST}$ is not defined. The maximum $RH_{SST}$ shifts during the event, from above 60 % before the most intense precipitation period to 55 % at 12 UTC on 07 Dec 2016. It appears that the most intense precipitation

stage was thus also associated with the most intense evaporation due to the strongest humidity gradient over the North Atlantic moisture sources.

For comparison with the stable isotope measurements, we predict the $d$-excess at the moisture source from the empirical relation between $RH_{SST}$ and $d$-excess by Pfahl and Sodemann (2014) (Fig. 8d). The highest $d$-excess from the moisture sources is predicted during the peak of the precipitation event, with a maximum at 16‰ (grey shading). As for $RH_{SST}$, land

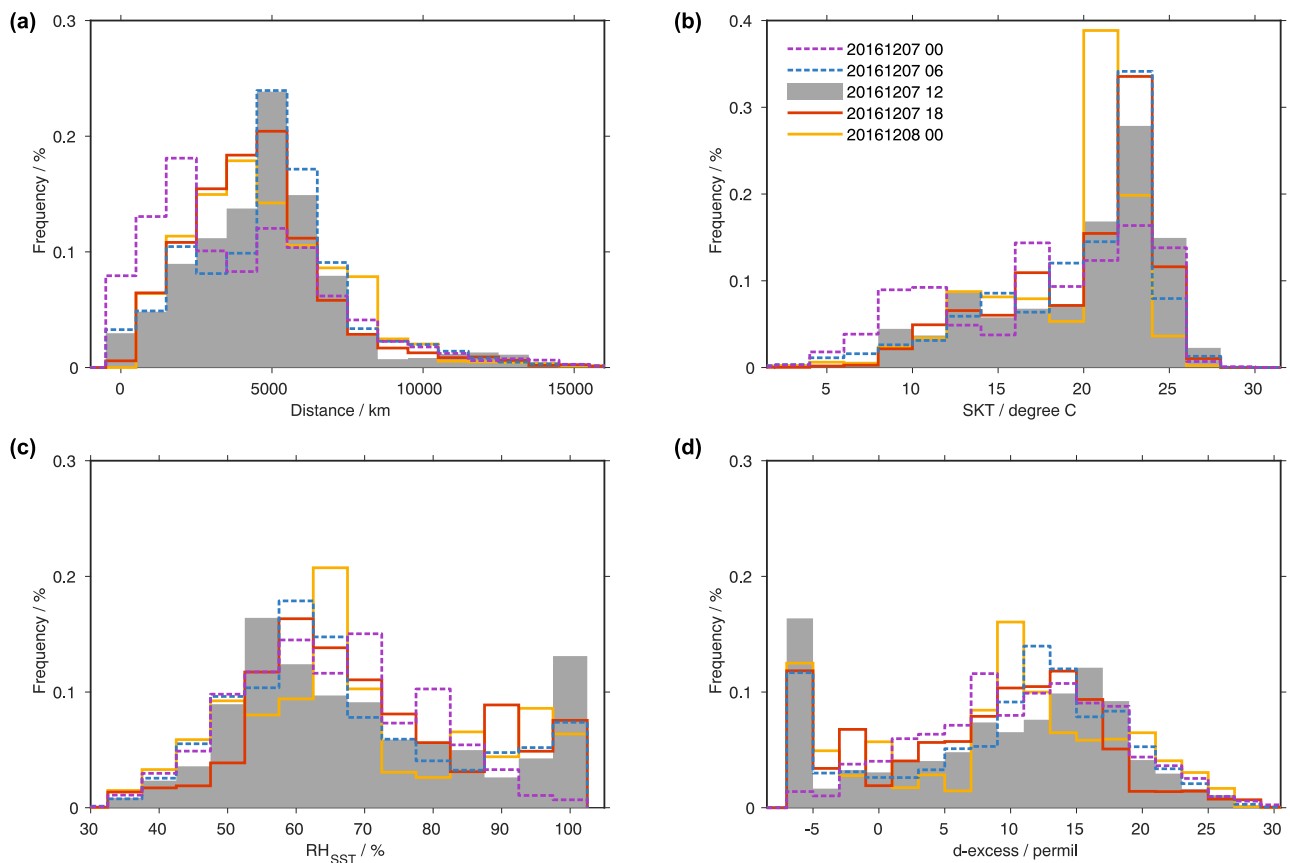

**Figure 8.** Histograms of moisture source conditions identified with the Lagrangian moisture source diagnostics from 20-day backward trajectories during the AR precipitation event in southwestern Norway. **(a)** Moisture source distance (km), **(b)** moisture source temperature (°C), **(c)** moisture source relative humidity with respect to sea surface temperature ($RH_{SST}$), and **(d)** $d$-excess estimated from the empirical relation of (Pfahl and Sodemann, 2014). Grey filled bars show the most intense period of the event (Phase III, 07 Dec 2016 at 12 UTC), solid lines the 12 h before (Stage I and II), and dotted lines the 12 h after the central period (Stage IV). Histograms represent the normalized contributions of each moisture source location to the precipitation at the arrival region on a respective date.




sources produce an artefact for $d$-excess below $-5\,‰$. Both before and after the main precipitation period, the maximum in the $d$-excess distribution is shifted to lower values. This sequence from low to high to low $d$-excess throughout the event is qualitatively consistent with the observed $d$-excess signal. The initial low and even negative $d$-excess in precipitation during Stage I is thus likely a combination of the moisture source conditions, amplified by below-cloud evaporation. The source $d$-
excess is more sensitive to $\mathrm{RH}_{SST}$ than to SST (Merlivat and Jouzel, 1979; Aemisegger et al., 2014). Considering additionally that the source temperatures only change slightly during the event, the humidity gradient above the moisture sources appears as the dominant driver of the $d$-excess changes observed here.

Considering a longer time period around the case investigated here, the Lagrangian diagnostic indicates a rather constant $d$-excess value during the whole precipitation event (Fig. C1d). The observed $d$-excess variation is not captured by the Lagrangian
diagnostic. The detailed inspection of Fig. 8 indicates the lack of variability is likely due to averaging the complex histograms to one value at the arrival location. The key characteristic of the histogram distribution is the maximum probability, but skewed and bimodal distributions make it difficult to provide more robust statistic measures. To represent the full variability of the moisture source conditions, detailed inspection of the moisture source properties throughout the event is therefore needed.

## 6  Discussion

We now return to the initially mentioned dispute in the literature regarding the interpretation of the precipitation isotope signal from an AR case making landfall at the coast of California. From sampling precipitation at a 30 min time interval during the AR event, C08 found a remarkable variation in $\delta$D of 60 ‰, progressing from less depleted to depleted and back. Both the shape and amplitude of the stable isotope variation were similar to the case studied here. C08 based the interpretation of the variability primarily on changes in cloud height, i.e. the temperature of condensation (Scholl et al., 2007). Using a
Rayleigh distillation model, C08 proposed that the initial phase precipitation would originate from low clouds with an average condensation temperature ($T_c$) of $10.0\,°C$, followed by deeper clouds with an average $T_c$ of $-4.2\,°C$, and again shallow clouds with $T_c$ of $9.7\,°C$.

Y10 then simulated the same AR event with a regional isotope-enabled model, leading them to propose a fundamentally different explanation for the isotope variation in surface precipitation observed by C08. According to that interpretation, the
initial drop from less depleted to depleted precipitation would be caused by below-cloud evaporation. Furthermore, Y10 found from their simulation that up to one-third of the condensate would be contributed from the lower troposphere (below 800 hPa), with an increasing tendency throughout the event. Notably, the contribution from the cloud top would decrease during the most depleted phase of the event. Despite uncertainties in some model parameters and parameterisations, Y10 concluded from their analysis that cloud microphysics, below-cloud exchange and advection all play a role for the observed isotope variation during
different phases of the event.

Expanding the dataset to 43 events sampled with a network of automatic rain samplers across northern California, Coplen et al. (2015) (henceforth C15) confirmed the pronounced isotope variation during events as seen in the case discussed in C08.



C15 argue that if the below-cloud kinetic exchange were to explain the initial enrichment in C08 as proposed by Y10, kinetic effects due to evaporation should have led to characteristic deviations from the GMWL.

The above controversy revolves around two questions: (i) What is the contribution from below-cloud interaction, and in particular evaporation, to the precipitation isotope signal? (ii) Are Rayleigh-type models adequate to explain the surface pre-
cipitation signal during AR cases? Based on our highly detailed analysis of an AR event, with high-resolution precipitation sampling and simultaneous water vapour measurements, we are in a situation to contribute constructively to both aspects of this scientific controversy.

## 6.1   Contribution from below-cloud interaction to the isotope composition in surface precipitation

Y10 proposed that below-cloud processes can explain the isotopic enrichment in precipitation observed at the beginning of
the C08 event, rather than cloud height. The joint observation of both surface vapour and precipitation in this study shows a characteristic time lag of the vapour over the precipitation signal. One plausible explanation for this time lag is that diffusional interaction takes place between precipitation and the surrounding vapour over extended time periods. Even though the total column mass of precipitation in a column is typically only about $1/10^{th}$ of the IWV, precipitation persisting over longer periods will imprint on ambient vapour isotope composition, and vice versa. As more precipitation falls, the below-cloud air gradually
saturates, reduce the vertical isotope gradient, and eventually reach isotopic equilibrium with the precipitation. At that point, the time lag between precipitation and vapour isotopes would vanish. Here, we find this time lag to be on the order of 30 min.

As long as the surface air is unsaturated, net mass transfer is directed away from raindrops, thus below-cloud evaporation reduces drop sizes and rainfall amounts, causing characteristic deviations in the $\Delta\delta\Delta D$ framework that reflect kinetic frac-tionation effects. The rainfall contributed during Stage I in this study was however too small to markedly influence the isotope
composition of the rainfall total (Table 1). Concerning the scientific controversy introduced above, we note that below-cloud processes can influence precipitation and surface vapour, but that the signal can be too small to detect if samples are too long, or due to sampling and analytical uncertainty. It is therefore not possible to confirm that the initial enrichment in C08 dataset was actually due to below-cloud evaporation, in particular without additional vapour measurements. Other factors, such as advection effects or progressive vapour/precipitation exchange could also have contributed to the initial enrichment.

## 25  6.2   Adequacy of the Rayleigh model to explain the isotope composition in surface precipitation

The majority of the precipitation in ARs is arriving with the strong onshore flow of the warm sector, led by the warm front and dominated by long-range transport. Large-scale ascent, enforced by orographic lifting and condensation heating during landfall leads to condensation and predominantly stratiform cloud formation. The warm conveyor belt (WCB) model is often used to describe the strongest precipitation-generating airflow in the warm sector of cyclones (Madonna, 2013). According
to a common classification criterion, airmasses in the WCB airstream rise $300\,\mathrm{hPa}$ or more in $48\,\mathrm{h}$, corresponding to vertical ascent on the order of several $\mathrm{cm\,s^{-1}}$. Precipitation from cold-sector airmasses, in contrast, has a more convective nature, characterized by an isolated ascent in updrafts, and dominated by vertical motions on the order of up to several $\mathrm{m\,s^{-1}}$.





From the Rayleigh model simulation presented in Sect.5.2, we find that the condensation temperature of the surface precipitation is most consistent with the temperature profiles in the reanalysis data (Fig. 6, purple contours) when interpreted as a representation of the vapour-mass-weighted average in the column rather than the cloud base or cloud top temperatures. MRR2 reflectivity profiles for the four precipitation stages considered here confirm that lower levels contribute substantially to the
surface precipitation.

Variants of the Rayleigh distillation model are often used to represent the isotope fractionation during condensation processes (e.g. Jouzel and Merlivat, 1984). In nature, however, precipitation will enter from above into subsequent air parcels from below. This process, as well as the isotopic exchange of the falling precipitation with air parcel vapour (in case of liquid phase), is not part of Rayleigh distillation model. Indeed, the Rayleigh model may thus only be adequate to simulate the vapour composition
in a rising air parcel, and the precipitation falling directly from it, which can be adequate for some convective-type precipitation processes. In the case of the more slowly ascending warm-sector airmasses, however, where clouds contribute to condensation at a range of atmospheric layers, a single air parcel appears insufficient to capture the actual precipitation process. Conceptually, it could be possible to consider instead an entire stack of Rayleigh-model air parcels as a better representation of stratiform cloud processes. Each air parcel in the column is at or near saturation, contains cloud droplets, and will receive input of
hydrometeors from above. Each air parcel will thus contribute to the precipitation by condensation or deposition, riming, scavenging, and partially equilibrate with the water vapour on passing through. The vertical connection of an entire stack of Rayleigh-type parcels creates a more efficient, coupled fractionation process than an isolated Rayleigh-type parcel as in the convective case. Given such a vertically coupled perspective, a single cloud top or condensation temperature from one Rayleigh process appears too limited to capture the influences on the fractionation process in the entire cloud. This is underlined by the
fact that the Rayleigh model used in C08 only needed temperatures down to $-4.2$ °C to explain the observed precipitation isotopes, which could not be reconciled by the range of temperatures throughout the entire column found by Y10. A similar observation was made here with the Rayleigh model of Jouzel and Merlivat (1984).

As the precipitating warm-frontal airmass is advected horizontally with the AR, it will produce a coherent isotopic signal at the surface, as noted by the displacement times in C15. C15 also noted that there is no immediate relation between the isotopic
depletion and either the total amount or the intensity of precipitation during landfall. Both of these findings are consistent with the interpretation that the isotope composition of the stratiform cloud can obtain a coherent, depleted isotope signature from a sustained lifting process. The isotopic signal of stratiform cloud then reflects a time-integrated condensation history of the airmasses, whereas surface precipitation is a combination of the airmass signature, the surface vapour, and the below-cloud interaction processes.

We conclude from this discussion that since the isotopic precipitation signal is intimately coupled to the cloud microphysics and dynamics, the Rayleigh perspective can be adequate to represent the isotope composition near cloud top and in some convective situations. However, for surface precipitation, and precipitation from deep stratiform clouds in frontal systems, such as ARs, the Rayleigh model reaches conceptual limitations. Despite their own uncertainties, it, therefore, appears necessary to invoke more complex numerical tools in the interpretation, such as isotope-enabled numerical weather prediction models, or
Rayleigh-type models adapted to stratiform clouds.





## 7    Conclusions and further remarks

We have presented a high-resolution stable isotope signature of a land-falling atmospheric river event in southwestern Norway during winter 2016. Figure 9 provides a conceptual summary of the sequence of events, by providing a spatial depiction of the airmasses arriving at Bergen. In surface precipitation, we observe $\delta$D that develops in a stretched "W" shape (between $-180$

and $-100$ ‰ for equilibrium vapour of precipitation), and $d$-excess that increases from $-9$ to 13 ‰, followed by a gradual decrease to 0 ‰. In surface vapour, $\delta$D exhibits the same "W" shape, following closely to the precipitation isotope variation, with a lag of about 30 min. The $d$-excess in vapour differs in the beginning markedly from precipitation signal, increasing from 10 to 16 ‰. As relative humidity below cloud base increases, the vapour $d$-excess follows the same trend as that of precipitation, reaching 0 ‰ at the end of the event.

Combining isotope and meteorological observations, we have identified four different precipitation stages during the event. At each stage, weather-system processes imprint on the isotope variations (Fig 9). Specifically, at the beginning of the event (Stage I), below-cloud evaporation is substantial, contributing to the low and even negative $d$-excess and relatively enriched $\delta$D in surface precipitation. At Stage II, the gradual weakening of below-cloud evaporation as ambient air becomes more saturated, and the involvement of hydrometeors from above the melting layer results in a gradual drop of $\delta$D and an increase in $d$-excess.

At Stage III, deep clouds allow hydrometeors formed at high levels to gain moisture from low levels, leading to intermediately depleted values in $\delta$D. Stage IV is characterised by numerous convective showers that are formed at relatively low elevation, leading to the most enriched $\delta$D values during the event. The gradual drop of the $d$-excess in both surface precipitation and vapour during Stage III and IV can at least partly be explained by a change in moisture source conditions.

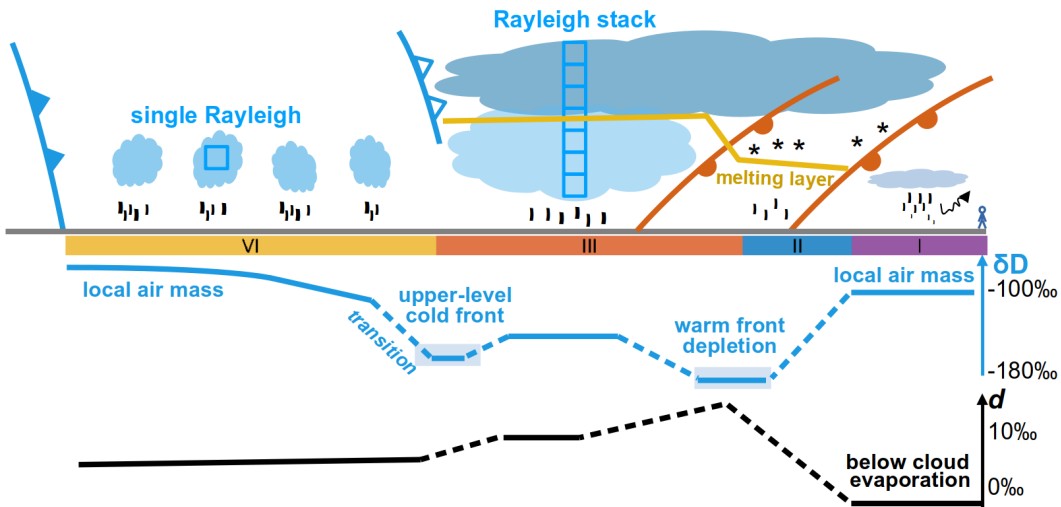

**Figure 9.** Weather diagram for the precipitation event at Bergen on 07 December 2016. $\delta$D and $d$-excess lines represent the evolution of isotope composition of surface vapour or equilibrium vapour from surface precipitation. Precipitation periods (I, II, III and IV) are indicated with color bars at the bottom. isotope Note that the timeline is from right to left.





Regarding the controversial discussion of the isotopic signal during previous AR events in the literature (C08, Y10, C15), we emphasize from our results that the isotopic precipitation signal is intimately coupled to the cloud microphysics and dynamics. Idealized Rayleigh models may be adequate to represent the isotope composition of water vapour near cloud top during convective precipitation events. However, additional factors and more complex models should be considered to interpret the
isotopic signal in surface precipitation, in particular for deep, stratiform clouds. A stack of Rayleigh models could be a more adequate conceptual view for these cloud types (Fig 9).

Our case study provides a unique isotope dataset of an AR event in southwestern Norway. More cases should be performed in the future to test the more general validity of the results obtained in this case study. However, from one case already it is apparent that the isotopic information from combined (paired) water vapour and precipitation isotope sampling can be highly
valuable for future data-model comparison studies with isotope-enabled weather prediction models.

*Data availability.* Datasets are available in the supplement.

## Appendix A: Comparison of precipitation rate measurement

The precipitation rate at the sampling site (45 m a.s.l.) is measured by two instruments, i.e. Total Precipitation Sensor (TPS-3100) and Parsivel[2] distrometer. Fig. A1 shows a comparison of hourly precipitation rate during the precipitation period
between the measurements of these two instruments and that of the rain gauge measurement from the closest meteorological station (70 m away, 12 m a.s.l.). The comparison shows that the TPS-3100 measures a slightly higher precipitation rate while the Parsivel[2] recorded a substantially lower precipitation rate, particularly in the situation of heavy precipitation. Since the TPS-3100 measurements agree well with the rain gauge measurements, we choose to use the precipitation rate from TPS-3100 for the analysis in this study. We did not choose to calibrate the TPS-3100 measurements against the rain gauge measurements
because the small discrepancy can be due to the different locations and elevations of the two instruments.

## Appendix B: Sensitivity studies with the Below-Cloud Interaction Model (BCIM)

Idealized simulations with the BCIM model help to reveal the sensitivity to factors influencing the below cloud processes. The background profiles of the BCIM model were here obtained from the moist adiabatic ascent of an air parcel that is lifted from the surface with initial values of $T_0 = 5\,°C$ and $RH_0 = 90\,\%$. The background isotope profiles are obtained correspondingly
from Rayleigh fractionation with a surface composition of $\delta D = -160\,‰$ and $d$-excess $= 10\,‰$. A formation height of $1500\,m$ was used in this reference simulation. Other simulations are obtained in the same way with an adiabatic ascent of an air parcel, while modifying one of the initial values as detailed below:

– The sensitivity to RH was evaluated by modifying the surface RH in steps of 2 % between 64 and 100 % while keeping all other parameters unchanged.





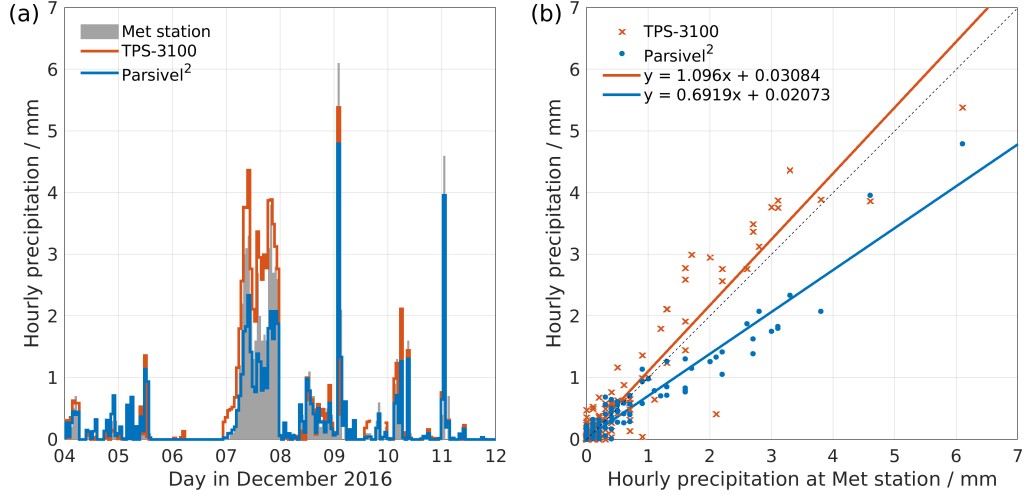

**Figure A1. (a)** Hourly precipitation rate during 04-11 December 2016 measured by rain gauge at WMO station (shading), TPS-3100 (red) and Parsivel[2] (blue). **(b)** Scatter plots and corresponding fits for the measurements of TPS-3100 (red) and of Parsivel[2] (blue) against those from rain gauge at WMO station.

- The sensitivity to formation height was evaluated by modifying the formation height in steps of 250 from 500 m to 3000 m while keeping all other parameters unchanged.

- The sensitivity to the temperature profile was evaluated by modifying the surface temperature in steps of 1 °C while keeping all other parameters unchanged.

5    While BCIM provides helpful insights, its limitation should be noted. The model only considers a single falling hydrometeor and assumes that the background isotope profile of the atmosphere is not affected by evaporating hydrometeor or other processes during the simulation. However, in our AR case presented here, it can be clearly seen that the precipitation has a profound influence on the isotopic evolution of surface vapour.

The BCIM is available from the website https://git.app.uib.no/Harald.Sodemann/bcim (see Graf et al. (2019) for more de-
10    tails).

**Appendix C: Long term observations and Lagrangian diagnostics**

To examine our AR precipitation event in the context of the longer-term weather evolution, we present here selected observations at the sampling site as well as the Lagrangian moisture source diagnostics for the Bergen region between 04 and 11 December 2016 (Fig. C1).

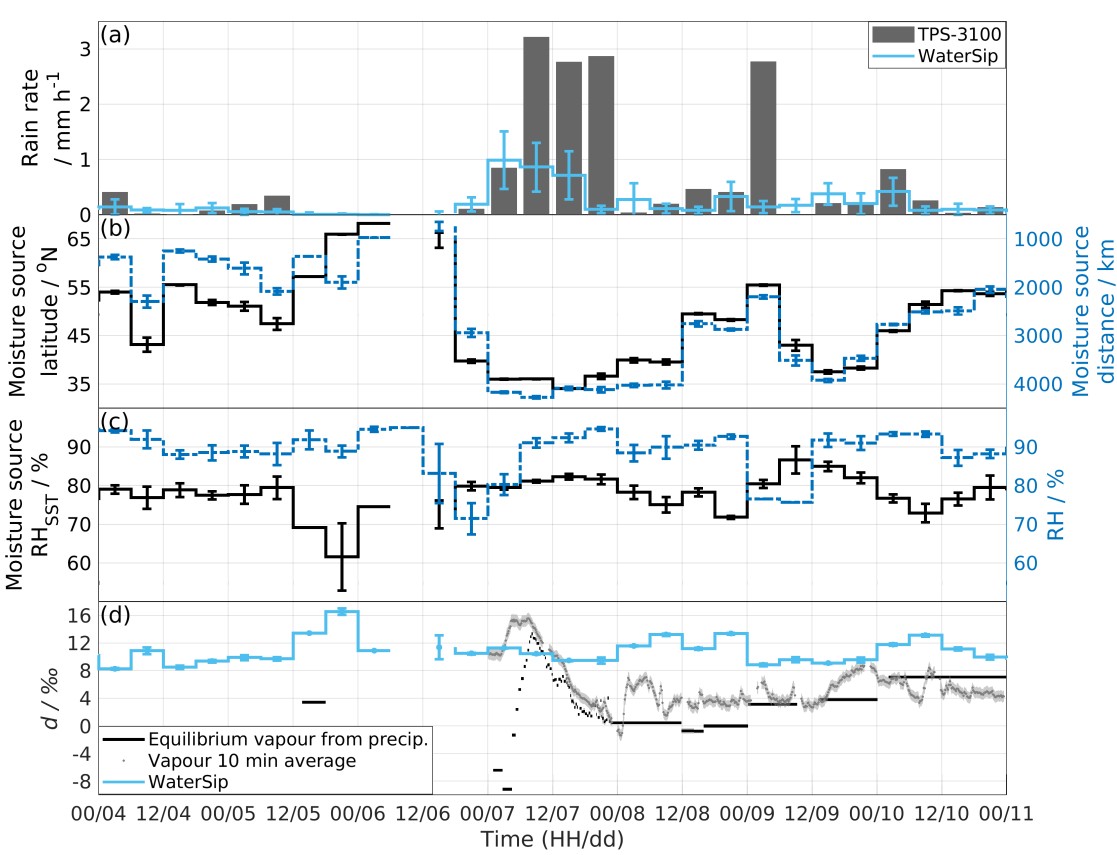

**Figure C1.** Seven days time series of observations at sampling site and Lagrangian diagnostic (WaterSip) output for the Bergen region between 00 UTC 04 December and 00 UTC 11 December 2016. **(a)** 6 hourly averaged rain rate observed from Total Precipitation Sensor (blue line) and estimated rain rate from WaterSip (grey shading). **(b)** Moisture source latitude (solid black line) and source distance (dashed blue line) estimated by WaterSip. **(c)** Moisture source $RH_{SST}$ (solid black line) estimated by WaterSip and 6 hourly averaged RH at sampling site (dashed blue line). **(d)** $d$-excess of the 10 min averaged vapour (grey dots), of the equilibrium vapour from precipitation (black segments) at 45 m above ground, and WaterSip estimate (light blue). The width of the black segment indicates the period over which the precipitation sample was collected. The uncertainties are 0.83 ‰ and 0.20 ‰ for $d$-excess of vapour and of the equilibrium vapour from precipitation, respectively. The error bars in **(a-d)** indicate one standard deviation. The missing data of the WaterSip on 12 UTC 6 December is due to bad data quality. The observation of $d$-excess is only available from 07 December.





A dry period of one and a half day precedes the AR precipitation event. Following the AR precipitation, discontinuous, moderate precipitation occurs (Fig. C1a). The comparison of the precipitation time series shows a qualitative agreement, but with substantially lower precipitation intensities estimated by the Lagrangian diagnostic. The discrepancy in the precipitation intensity likely arises from the neglect of microphysical processes in the trajectory-based diagnostic, and from the limitation

of comparing a regional estimate with a single-point ground observation. The Lagrangian moisture source diagnostic shows that the dominating moisture source for the dry period pre the AR precipitation came from the north of Bergen (N of 65 °N; Fig. C1b, black solid line). During the AR precipitation, the moisture source shifted markedly to the south, reaching 35 °N. After the AR event, the moisture source gradually shifts back to the north, reaching 55 °N on 9 December, followed by another south-to-north variation. Closely following the source latitude, the moisture source distance reveals the airmass evolution from

a local airmass pre AR event, to a substantial remote airmass during the AR event, and a moderate-distance airmass after the AR event (Fig. C1b, blue dashed line). The estimated $RH_{SST}$ at moisture source indicates relatively intense evaporation condition at the moisture source before the AR event ($RH_{SST}$ reaching 62 %), more moderate evaporation condition during the AR event ($RH_{SST} \approx 80$ %), and varying evaporation conditions afterwards ($RH_{SST}$ varying between 72 and 85 %; Fig. C1c, black solid line). The local RH at the sampling site stays high (above 90 %) during the entire period, except at the beginning

of the AR event and between UTC 00 and 12 on 9th December (Fig. C1c, blue dashed line).

Finally, we examine the $d$-excess of near-surface vapour, of equilibrium vapour from precipitation, and the $d$-excess estimation based on Lagrangian diagnostics (Fig. C1d). The $d$-excess of surface vapour exhibits a peak (above 8 ‰, with a maximum of about 16 ‰) during the first half-day of the AR event. Thereafter, the $d$-excess of surface vapour remains at low levels mostly between 0 and 8 ‰. The low $d$-excess can be due to the calm evaporation conditions at the moisture source or a contribution

from land regions. The $d$-excess of equilibrium vapour from precipitation follows the overall variation of $d$-excess of surface vapour. The lower $d$-excess values for the quasi-daily precipitation samples collected after the AR precipitation event can be due to below cloud evaporation, and cloud microphysical processes.

*Author contributions.*   YW and HS designed the study jointly. Observation and data analysis was led by YW with contributions from AJ and HS. All authors contributed to the writing of the paper.

*Competing interests.*   The authors declare that no competing interests are present in this study.

*Acknowledgements.*   We thank Ole Edvard Grov for managing and providing the data from Automatic Weather Station (AWS-2700) and Norwegian Met office for high-resolution precipitation data at the Bergen-Florida station. This research has been supported by the Research Council of Norway SNOWPACE (grant no. 262710) and FARLAB (grant no. 245907).




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
