# Peer review of "High-resolution stable isotope signature of a land-falling Atmospheric River in southern Norway"

_Weather and Climate Dynamics, 2020_

## Referee Comment (RC1) · Anonymous Referee #1 · 24 Dec 2020

This paper describes a detailed case study of an AR event that occurred in Norway. They investigated the causes of the observed W-shaped isotope evolution using various methods focusing on below cloud evaporation, advection/mixing of air masses due to frontal dynamics, and moisture source. They also explained the interpretations of isotope dynamics by past studies during the AR event in California. I think the content of the paper is beneficial to understand isotopic behavior in ARs, and it is worth publishing. However, the paper is overall very reader unfriendly. There are so many methods involved in this study with many acronyms and many redundant discussions.

I think we can condense the paper considerably and still be able to tell the key findings of this paper.

1. In the Introduction, first describe the background of the study and then list key research questions you would like to answer. The authors described precipitation -> isotope -> AR, and raised the first question of uncertain moisture source of AR in p2, line 18-19. Then, they went back to describe precipitation isotopes. The flow of the Introduction needs to be improved.

2. p3, line 10 – p4 line 11: This part is redundant and does not belong to the Introduction. It is mentioned later in my comment, but I think some of these citations need to appear in the Results section when you describe certain processes. Either list the key components in 1 or 2 lines, or remove this part.

3. Remove p4 line13-22. Describe here what are your research questions and briefly explain how you are going to approach them.

4. Remove p4 line 24-25. Not necessary.

5. p4 line28: what is sharki.oslo.dnmi.no?

6. p7-8: why so many datasets are used? Could it be analyzed just by ERA5? I do not get the rationale for using this many data.

7. Figure 1: just show either one. The aim of this study is not a verification of the satellite dataset. ARs can be simulated well enough in reanalysis for this study purpose.

8. Figure 2: Why do you use forecast data? Can it be from ERA5? Please consider reducing datasets for simplicity.

9. Section 3.1 and Section 4: These parts are very redundant and painful to read. Some of the explanations appear many times when explaining different variables. The important thing in this part is to list key behaviors that you are going to investigate in Section 5. You do not need to list all behaviors shown in Figure 3. Therefore, I suggest

rewriting it in the following format. First, describe how you separate the event into 4 stages based on frontal conditions. Then, describe key behaviors in each stage (I -> IV). The content in sections 3.1 and 4 can be combined.

10. p14 line14: Are there any common features in the past W-shape event? Such as frontal dynamics?

11. Section 5: In this section, I think it would be more accessible to add some more citations when interpreting results.

12. p15 line28-29: "does not yet..." I do not get it. Please make it more accessible. Also, I understand that positive $\Delta\delta$ indicates strong re-evaporation, but what does "evaporation is incomplete" mean when it's negative?

13. Figure 5a: Could you change the shape of the markers according to each stage? It isn't easy to memorize when each stage starts so I think it's helpful for readers.

14. Figure 5b-d: I think it is helpful to use discrete colormap levels rather than continue colormap since you refer to some exact numbers in the text.

15: Figure 5: I do not understand how each line is calculated. For example, in Figure 5b, which parameter is modified? RH?

16: Figure 5: It seems to me that overall RH, precipitation, droplet diameter only explain $\Delta$d but not $\Delta\delta$. Droplet size seems to explain $\Delta\delta$ variability to some extent but not so clear. What could be the reasons?

17: p17, line13-14: "coordinate system of drop-size dependent effects of RH on raindrops"? Please explain.

18: p18 line 2-3: more HDO is transferred to the liquid phase?

19: p19 line1: "This indicates the influence of the below-cloud exchange." Is this true? I thought $\Delta$d indicate the amounts of below-cloud evaporation. Some citations might help.

20: p20 line18-19: Could you make it more accessible?

21: p20 line22 and throughout: Is there any difference in meaning between "less depleted" and "enriched"? Can we just use "enriched"?

22: p20 Line25: cite some papers on heating profiles and isotopes

23: p20 line29 – p22 line3: I think we can remove this part or move it to supplement. I do not see any key messages in this part. Aside, Table 1 should be a figure to compare simulation and observations.

24: p22 line10 – p24 line6: This part overlaps with section 3 so make it short and combine with section 3. The key message here is that the moisture uptake region coincides with AR tracks. The right side of figure 7 is not necessary and distracting.

25: Figure 8: the colors of the first two dashed lines are a bit difficult to distinguish. Could you change one of the colors?

26: Figure 8: Could you also make figures of mean or medians of these pdfs since that is mainly discussed in the text?

27: p28, line 34: If I understand correctly, Y10 uses isoRSM. Based on your discussion, can we consider the claims of Y10 are more reliable?

28: Abstract line16-18: Make it more accessible to readers who only read the abstract.

29: Abstract line 20: "AR events in California"

30: Abstract line 20-25: This part may not be suitable for the abstract. The conclusions are too ambiguous.

---

## Referee Comment (RC2) · Anonymous Referee #2 · 5 Jan 2021

The research presented by Weng et al. deals in detail with winter 2016 AR event in Norway. Although the study is really interesting, some parts are a bit hard to read. This is especially obvious in parts of introduction, but as well in discussion. In general, discussion part of the paper is lacking references. As well, references should be written properly and not using acronyms.

- p3, line 10 – p4 line 11: This part does not belong to the Introduction and should be moved to later part of the manuscript. - p4 line 13-22: This part should be rewritten in more concise way. - p4 line 28: the reference to web page should be written differently and should be listed properly in list of references. Same goes for P4 line 32 and P8 line

3. - p9 fig 1. I believe panel b would be sufficient since you are using ERA-Interm data. Thus, in p8 lines 4-5 are not necessary. - p21 (Table 1 and line 3-16) it would be worthy to put observed data in the table to enable easier comparison. Doing so, this part of the text can be significantly reduced or even eliminated. - Fig 8. It is really hard to read it. Legend should be set to be representative for all panels (this way looks that is valid only for b panel). As well, please check solid and dotted lines in figure description are designated properly. - p27, line 24. Other factors - p 28, line 7 ???

---

## Author Comment (AC1) · 23 Feb 2021

We are grateful for both reviewer's helpful comments that help us to clarify several aspects in the manuscript and improve the presentation quality.

**Anonymous Referee #1**

This paper describes a detailed case study of an AR event that occurred in Norway. They investigated the causes of the observed W-shaped isotope evolution using various methods focusing on below cloud evaporation, advection/mixing of air masses due to frontal dynamics, and moisture source. They also explained the interpretations of isotope dynamics by past studies during the AR event in California. I think the content of the paper is beneficial to understand isotopic behavior in ARs, and it is worth publishing. However, the paper is overall very reader unfriendly. There are so many methods involved in this study with many acronyms and many redundant discussions.

I think we can condense the paper considerably and still be able to tell the key findings of this paper.

1. In the Introduction, first describe the background of the study and then list key research questions you would like to answer. The authors described precipitation -> isotope -> AR, and raised the first question of uncertain moisture source of AR in p2, line 18-19. Then, they went back to describe precipitation isotopes. The flow of the Introduction needs to be improved.

We will rephrase and shorten the introduction. In particular, exchanging the sequence of the first and second paragraph will improve the flow of the introduction.

2. p3, line 10 – p4 line 11: This part is redundant and does not belong to the Introduction. It is mentioned later in my comment, but I think some of these citations need to appear in the Results section when you describe certain processes. Either list the key components in 1 or 2 lines, or remove this part.

We agree that this part can be shortened in the Introduction and corresponding references included in the results. We will replace this section by a more general description of the threefold approach focusing on moisture sources, moisture transport, and post-condensation effects.

3. Remove p4 line13-22. Describe here what are your research questions and briefly explain how you are going to approach them.

Our research questions and approach are explained in the sections p3, line 10 to p4 line 11 that the reviewer proposes to shorten and move. This section will be adjusted in the light of our revision to these sections.

4. Remove p4 line 24-25. Not necessary.

We consider this a matter of style, and prefer to keeping this sentence, since it can provide a better flow of the methods section to some of our readers.

5. p4 line28: what is sharki.oslo.dnmi.no?

This is a data api for meteorological observations archived with the Norwegian Meteorological Institute. This will be clarified in an additional sentence.

6. p7-8: why so many datasets are used? Could it be analyzed just by ERA5? I do not get the rationale for using this many data.

We agree that the rationale for using several meteorological datasets needs to be given. In brief, we used ERA-Interim as basis for the moisture source diagnostics and meteorological large-scale figures. ERA5 reanalysis provides hydrometeors at a higher time resolution, and was therefore used to produce Fig. 6. The regional operational forecast model data is available at much higher resolution than ERA5 or ERA-Interim in the vicinity of our measurement site, and provides a more detailed view of the frontal passages. We think that with a more detailed explanation of the rationale, it will become clear that the use of several data sources is a strength rather than a weakness of our study. The description of satellite data will be removed from this section.

7. Figure 1: just show either one. The aim of this study is not a verification of the satellite dataset. ARs can be simulated well enough in reanalysis for this study purpose.

We will remove the satellite image and corresponding description from the manuscript, and replace it with a short descriptive sentence.

8. Figure 2: Why do you use forecast data? Can it be from ERA5? Please consider reducing datasets for simplicity.

We use the much higher resolution model with a limited domain for depicting the approaching frontal system (see comment #6). In particular, the local precipitation patterns benefit from using the high-resolution forecast dataset.

9. Section 3.1 and Section 4: These parts are very redundant and painful to read. Some of the explanations appear many times when explaining different variables. The important thing in this part is to list key behaviors that you are going to investigate in Section 5. You do not need to list all behaviors shown in Figure 3. Therefore, I suggest rewriting it in the following format. First, describe how you separate the event into 4 stages based on frontal conditions. Then, describe key behaviors in each stage (I -> IV). The content in sections 3.1 and 4 can be combined.

In our revision, we will rephrase sections 3.1 and 4, and consider combining them, following the reviewer's suggestion.

10. p14 line14: Are there any common features in the past W-shape event? Such as frontal dynamics?

We will include a more detailed comparison to the interpretations in Muller et al., 2015.

11. Section 5: In this section, I think it would be more accessible to add some more citations when interpreting results.

We will incorporate some of the citations from the introduction here, that will be removed according to comment #3.

12. p15 line28-29: "does not yet..." I do not get it. Please make it more accessible. Also, I understand that positive Δδ indicates strong re-evaporation, but what does "evaporation is incomplete" mean when it's negative?

This is a typing error, and will be rephrased as «below-cloud equilibration is incomplete».

13. Figure 5a: Could you change the shape of the markers according to each stage? It isn't easy to memorize when each stage starts so I think it's helpful for readers.

This is a good suggestion and we will include this in the revised Fig. 5.

14. Figure 5b-d: I think it is helpful to use discrete colormap levels rather than continue colormap since you refer to some exact numbers in the text.

The colormap of the symbols is in fact discrete, but the colorbars showed up as continuous in the print. This will be fixed in the revised manuscript.

15: Figure 5: I do not understand how each line is calculated. For example, in Figure 5b, which parameter is modified? RH?

The sensitivity experiments are described in appendix B. We will refer to this appendix from the figure caption, and improve the explanation of the different sensitivity experiments in the appendix. Regarding RH, the surface value was modified in the range from 64 to 100%, and then interpolated linearly to 100% at cloud based height.

16: Figure 5: It seems to me that overall RH, precipitation, droplet diameter only explain $\Delta d$ but not $\Delta \delta$. Droplet size seems to explain $\Delta \delta$ variability to some extent but not so clear. What could be the reasons?

For the first phase, indicated by letter A, the $\Delta \delta$ and $\Delta d$ are anti-correlated regarding RR and RH. A possible explanation is, that the background vapour profile has a strong impact on the $\Delta \delta$, such that it dominates over below-cloud effects. We did not change the background isotope profile as part of our sensitivity tests. The time offset (about half an hour) between the surface vapour and precipitation isotope signals also plays a role here. A brief discussion according to comment #16 will be included in the revised manuscript.

17: p17, line13-14: "coordinate system of drop-size dependent effects of RH on raindrops"? Please explain.

The sentence intends to present results of the sensitivity experiments plotted as lines in panels a-d as a coordinate system, that allows to identify the influence of different parameters, such as RH from the location of data points. We will rephrase this sentence for clarity in the revised manuscript.

18: p18 line 2-3: more HDO is transferred to the liquid phase?

This will be rephrased into two sentences for clarity. More $H_2^{16}O$ leaves the liquid during evaporation than light isotopes due to fractionation. However, due to non-equilibrium conditions, relatively more HDO than $H_2^{18}O$ will leave the droplet.

19: p19 line1: "This indicates the influence of the below-cloud exchange." Is this true? I thought $\Delta d$ indicate the amounts of below-cloud evaporation. Some citations might help.

We use the term below-cloud exchange as an overarching term for both below-cloud evaporation and below-cloud equilibration, and yes, below-cloud evaporation dominates the $\Delta d$. We will clarify this in Sec. 5 and carefully check for when to specifically use the term below-cloud evaporation.

20: p20 line18-19: Could you make it more accessible?

We will explain with an additional sentence how the upper-level front contributes to an excursion in the precipitation isotope signature.

21: p20 line22 and throughout: Is there any difference in meaning between "less depleted" and "enriched"? Can we just use "enriched"?

We will use a more consistent language to denote relative isotopic composition throughout the manuscript.

22: p20 Line25: cite some papers on heating profiles and isotopes

We will include more references in this paragraph.

23: p20 line29 – p22 line3: I think we can remove this part or move it to supplement. I do not see any key messages in this part. Aside, Table 1 should be a figure to compare simulation and observations.

The intention and key message here is to show the deficiencies of using a Rayleigh model interpretation with derived cloud top heights to predict the d-excess value. Apparently we did not sufficiently motivate this analysis and highlight its importance (see reviewer 2, comment). This section will be improved in the revised manuscript.

24: p22 line10 – p24 line6: This part overlaps with section 3 so make it short and combine with section 3. The key message here is that the moisture uptake region coincides with AR tracks. The right side of figure 7 is not necessary and distracting.

From our perspective, the logic of the results on moisture sources will be less clear and potentially confusing when this section is combined with Sec. 3. Some overlap with section 3 can not be avoided in this section, and we will consider shortening where possible. We will also consider replacing the right side of Fig. 7 with displays of IWV.

25: Figure 8: the colors of the first two dashed lines are a bit difficult to distinguish. Could you change one of the colors?

We will change one of the colours to green.

26: Figure 8: Could you also make figures of mean or medians of these pdfs since that is mainly discussed in the text?

We will include indicators for the median in the revised Fig. 8.

27: p28, line 34: If I understand correctly, Y10 uses isoRSM. Based on your discussion, can we consider the claims of Y10 are more reliable?

We consider the approach of Y10 as more comprehensive, and therefore as potentially more reliable. However, comparison to paired vapour and precipitation measurements are in our view needed to back this expectation with evidence, as we note in the conclusions.

28: Abstract line16-18: Make it more accessible to readers who only read the abstract.

We will attempt to rephrase the sentence.

29: Abstract line 20: "AR events in California"

This wording will be revised accordingly.

30: Abstract line 20-25: This part may not be suitable for the abstract. The conclusions are too ambiguous.

We will consider rephrasing these conclusions, but note also that in our view the results obtained in Sec. 5.1 are sufficiently robust to issue a recommendation for careful interpretation.

**Anonymous Referee #2**

The research presented by Weng et al. deals in detail with winter 2016 AR event in Norway. Although the study is really interesting, some parts are a bit hard to read. This is especially obvious in parts of introduction, but as well in discussion. In general, discussion part of the paper is lacking references. As well, references should be written properly and not using acronyms.

1. p3, line 10 – p4 line 11: This part does not belong to the Introduction and should be moved to later part of the manuscript.

In line with the comment from reviewer 1, comment #2, this will be changed accordingly in the revised manuscript.

2. p4 line 13-22: This part should be rewritten in more concise way.

This section will be revised for conciseness.

3. p4 line 28: the reference to web page should be written differently and should be listed properly in list of references.

The reference to this website will be included according to journal style guidelines.

4. Same goes for P4 line 32 and P8 line 3.

OK, see comment above.

5. p9 fig 1. I believe panel b would be sufficient since you are using ERA-Interm data. Thus, in p8 lines 4-5 are not necessary.

We will remove the satellite data and according descriptions from the manuscript (see reviewer 1, comment #7)

5. p21 (Table 1 and line 3-16) it would be worthy to put observed data in the table to enable easier comparison. Doing so, this part of the text can be significantly reduced or even eliminated.

The observed data is already part of Table 1. In our opinion, text and table work well together, depending on reader preference. We will consider shortening part of the writing.

6. Fig 8. It is really hard to read it. Legend should be set to be representative for all panels (this way looks that is valid only for b panel). As well, please check solid and dotted lines in figure description are designated properly.

The legend will be moved to the outside of panel b, and lines will be offset for clarity. The figure caption will be corrected to be consistent with the legend.

7. p27, line 24. Other factors

Unfortunately, we could not understand this reviewer comment.

8. p 28, line 7 ???

Unfortunately, we could not understand this reviewer comment.

---

## Author Response (AR1)

We are grateful for both reviewer's helpful comments that help us to clarify several aspects of the manuscript and improve the presentation quality.

In addition to the point-by-point responses, we have carried out the following minor changes:

- changed the author's order, i.e. moved "Harald Sodemann" to the 3rd position
- separated the description for meteorological observations into a new section (Sect. 2.2)
- switched the position of Sect. 2.6 and Sect. 2.7
- changed the title of Sect. 3 from "Meteorological situation" to "Results"
- updated panel (d) of Fig. 3 using 1-min averaged MRR reflectivity data
- included the reference Rozanski and Sonntag (1982) in Sect. 5.2
- changed some words to keep the consistency throughout the paper, including some words changed from American English to British English

The detailed point-by-point response to all referee comments is provided below.

**Anonymous Referee #1**

*This paper describes a detailed case study of an AR event that occurred in Norway. They investigated the causes of the observed W-shaped isotope evolution using various methods focusing on below cloud evaporation, advection/mixing of air masses due to frontal dynamics, and moisture source. They also explained the interpretations of isotope dynamics by past studies during the AR event in California. I think the content of the paper is beneficial to understand isotopic behaviour in ARs, and it is worth publishing. However, the paper is overall very reader unfriendly. There are so many methods involved in this study with many acronyms and many redundant discussions.*

*I think we can condense the paper considerably and still be able to tell the key findings of this paper.*

1.  *In the Introduction, first describe the background of the study and then list key research questions you would like to answer. The authors described precipitation -> isotope -> AR, and raised the first question of uncertain moisture source of AR in p2, line 18-19. Then, they went back to describe precipitation isotopes. The flow of the Introduction needs to be improved.*

We exchanged the sequence of the first and second paragraph, which we think improved the flow of the introduction.

*2. p3, line 10 – p4 line 11: This part is redundant and does not belong to the Introduction. It is mentioned later in my comment, but I think some of these citations need to appear in the Results section when you describe certain processes. Either list the key components in 1 or 2 lines, or remove this part.*

We agree that this part can be shortened in the Introduction. We replaced this section with a more general description of the threefold approach focusing on moisture sources, moisture transport, and post-condensation effects. The more detailed paragraphs and corresponding references have now been included in the results:

"In order to disentangle different influences onto the isotope signal in precipitation, we consider three sets of factors together comprising the atmospheric water cycle of precipitation. Namely, these factors are: (1) ocean-atmosphere conditions at the moisture source that affect the isotopologue composition of generated water vapour, (2) the preferential loss of heavy isotopologues due to an atmospheric distillation or rainout process, (3) microphysical processes within clouds, and post-condensational exchange processes of falling precipitation that can alter the isotope composition. We hereby quantify the amount of isotopologues using the common delta notation as […], where […] is the isotope ratio, and the delta value quantifies the enrichment or depletion with respect to the Vienna Standard Mean Ocean Water standard (VSMOW)."

*3. Remove p4 line13-22. Describe here what are your research questions and briefly explain how you are going to approach them.*

Our research questions and approach are explained in the sections p3, line 10 to p4 line 11 that the reviewer proposed to shorten and move. This section has now been replaced by the following paragraph:

"In the following, we use a combination of atmospheric in-situ and remote sensing instrumentation at the measurement site, weather prediction model data, to identify periods in the sequence of the AR event where different factors have dominant or overlapping influences. To this end, we quantify below-cloud exchange processes by means of the interpretative ΔδΔD framework (Graf et al., 2019). We then relate the observed evolution of the isotope signal to the frontal structure and other weather system characteristics. Using the parameter d-excess, defined as d = δD −8·δ 18 O, and model diagnostics for moisture source location and evaporation condition analysis (Sodemann et al., 2008), we assess during which periods the precipitation isotope signal contains information about the evaporation conditions at the moisture sources. Based on the findings from our analysis, we attempt to resolve some of the disagreement between earlier observational and modelling studies of precipitation isotopes sampled at high resolution."

*4. Remove p4 line 24-25. Not necessary.*

Removed.

*5. p4 line28: what is sharki.oslo.dnmi.no?*

This is a data api for meteorological observations archived with the Norwegian Meteorological Institute. This has been clarified in an additional sentence:

"data retrieved from the observation data repository \url{https://sharki.oslo.dnmi.no}, Meteorologisk Institutt, Oslo, Norway"

*6. p7-8: why so many datasets are used? Could it be analyzed just by ERA5? I do not get the rationale for using this many data.*

We agree that the rationale for using several meteorological datasets needs to be given. In brief, we used ERA-Interim as a basis for the moisture source diagnostics and meteorological large-scale figures. ERA5 reanalysis provides hydrometeors at a higher time resolution, and was therefore used to produce Fig. 6. The regional operational forecast model data is available at a much higher resolution than ERA5 or ERA-Interim in the vicinity of our measurement site, and provides a more detailed view of the frontal passages. We have rephrased the paragraph to clarify that the use of several data sources is a strength rather than a weakness of our study. The description of satellite data has been removed from this section (we have also switched the position of this section and that of the following section):

"We use global ERA-Interim reanalysis data from the European Centre for Medium-Range Weather Forecast (ECMWF), re-gridded to a 0.75x0.75 degree regular grid, as the basis for the moisture source diagnostics, and for depicting the large-scale meteorological situation. Moisture transport is quantified by the integrated water vapour transport […], and mean sea level pressure (SLP) depicts the location of weather systems.

Due to the higher time resolution, vertical profiles of air temperature, solid and liquid precipitation, cloud water and cloud ice at the measurement site were extracted across all model levels from the ERA5 reanalysis with a 1-h time resolution. Finally, to depict the details of the frontal structure during the event, air temperature, horizontal wind speed and relative humidity at different pressure levels, as well as surface precipitation were obtained from high-resolution operational weather forecasts with the Harmonie-Arome model in the MetCoop domain. Forecasts initialized during the period 06 to 07 Dec 2016 at a grid spacing of 2.5x2.5 km were retrieved from the publicly accessible archive for weather forecast data (http://thredds.met.no, Meteorologisk Institutt, Oslo, Norway)."

*7. Figure 1: just show either one. The aim of this study is not a verification of the satellite dataset. ARs can be simulated well enough in reanalysis for this study purpose.*

We removed the satellite image and corresponding description from the manuscript, and replaced it with a short descriptive sentence.

*8. Figure 2: Why do you use forecast data? Can it be from ERA5? Please consider reducing datasets for simplicity.*

We use the much higher resolution model with a limited domain for depicting the approaching frontal system (see comment #6). In particular, the local precipitation patterns benefit from using the high-resolution forecast dataset. We have clarified the advantage of the much higher-resolution model data now in Section 2.5.

*9. Section 3.1 and Section 4: These parts are very redundant and painful to read. Some of the explanations appear many times when explaining different variables. The important thing in this part is to list key behaviors that you are going to investigate in Section 5. You do not need to list all behaviors shown in Figure 3. Therefore, I suggest rewriting it in the following format. First, describe how you separate the event into 4 stages based on frontal conditions. Then, describe key behaviors in each stage (I -> IV). The content in sections 3.1 and 4 can be combined.*

In our revision, we moved the stage description to the beginning of section 3.1, shortened section 3.1, and changed the section to subsection 3.2, following the reviewer's suggestion.

*10. p14 line14: Are there any common features in the past W-shape event? Such as frontal dynamics?*

The two W-shape events described in Muller et al. (2015) were obtained from surface observations only, and did not include vapour isotope measurements. While Muller et al (2015) speculate about a relation to frontal structure (cold/warm front alteration, double V-shape), we find that including a further discussion here would distract from the presentation of the results at hand here.

*11. Section 5: In this section, I think it would be more accessible to add some more citations when interpreting results.*

We incorporated some of the citations from the introduction here, that had been removed according to comment #3.

*12. p15 line28-29: "does not yet..." I do not get it. Please make it more accessible. Also, I understand that positive Δδ indicates strong re-evaporation, but what does "evaporation is incomplete" mean when it's negative?*

We have rephrased this explanation. Regarding the second remark, this is a typing error, and has been rephrased as «below-cloud equilibration is incomplete»:

"Negative Δδ values indicate that the (more depleted) isotope signal from the cloud level is preserved in precipitation, and has not been overprinted by below-cloud equilibration. In other words, below-cloud equilibration is incomplete in these cases."

*13. Figure 5a: Could you change the shape of the markers according to each stage? It isn't easy to memorize when each stage starts so I think it's helpful for readers.*

This is a good suggestion and we included this in the revised Fig. 5.

*14. Figure 5b-d: I think it is helpful to use discrete colormap levels rather than continue colormap since you refer to some exact numbers in the text.*

The colormap of the symbols is in fact discrete, the colorbars have been fixed in the revised manuscript.

*15: Figure 5: I do not understand how each line is calculated. For example, in Figure 5b, which parameter is modified? RH?*

The sensitivity experiments are described in appendix B. We now refer to this appendix from the figure caption, and improve the explanation of the different sensitivity experiments in the appendix. Regarding RH, the surface value was modified in the range from 64 to 100%, and then interpolated linearly to 100% at cloud base height. Each line is obtained from a range of drop sizes.

*16: Figure 5: It seems to me that overall RH, precipitation, droplet diameter only explain Δd but not Δδ. Droplet size seems to explain Δδ variability to some extent but not so clear. What could be the reasons?*

For the first phase, indicated by letter A, the Δδ and Δd are anti-correlated regarding RR and RH. A possible explanation is, that the background vapour profile has a strong impact on the Δδ, such that it dominates over below-cloud effects. We did not change the background isotope profile as part of our sensitivity tests. We included a brief discussion according to this comment in the revised manuscript when discussing Fig. 5.

*17: p17, line13-14: "coordinate system of drop-size dependent effects of RH on raindrops"? Please explain.*

The sentence intends to present results of the sensitivity experiments plotted as lines in panels a-d as a coordinate system, that allows to identify the influence of different parameters, such as RH from the location of data points. We rephrased this sentence for clarity in the revised manuscript.

*18: p18 line 2-3: more HDO is transferred to the liquid phase?*

More $H_2^{16}O$ leaves the liquid during evaporation than light isotopes due to fractionation. However, due to non-equilibrium conditions, relatively more HDO than $H_2^{18}O$ will leave the droplet. This has been rephrased for clarity.

*19: p19 line1: "This indicates the influence of the below-cloud exchange." Is this true? I thought Δd indicate the amounts of below-cloud evaporation. Some citations might help.*

We use the term below-cloud exchange as an overarching term for both below-cloud evaporation and below-cloud equilibration, and yes, below-cloud evaporation dominates the Δd in this case. This was rephrased, and we carefully checked for when to specifically use the term below-cloud evaporation versus the more general term below-cloud exchange.

*20: p20 line18-19: Could you make it more accessible?*

We explain by rephrasing the sentence how the upper-level front contributes to an excursion in the precipitation isotope signature:

"The very intense precipitation lasting for a 1-h period at the end of Stage III, associated with strong deviations in the Δδ-Δd-diagram, could be related to moist convection forming at this thermodynamic instability."

*21: p20 line 22 and throughout: Is there any difference in meaning between "less depleted" and "enriched"? Can we just use "enriched"?*

We now use a more consistent language to denote relative isotopic composition throughout the manuscript, using the terms "more/less depleted (with respect to VS-MOW)" throughout.

*22: p20 Line25: cite some papers on heating profiles and isotopes*

We now cite the study of Loewenthal et al. (2011) in this paragraph.

*23: p20 line29 – p22 line3: I think we can remove this part or move it to supplement. I do not see any key messages in this part. Aside, Table 1 should be a figure to compare simulation and observations.*

The intention and key message here is to show the deficiencies of using a Rayleigh model interpretation with derived cloud top heights to predict the d-excess value. Apparently, we did not sufficiently motivate this analysis and highlight its importance (see reviewer 2, comment). This section has been shortened and motivated more clearly in the revised manuscript. We found it difficult to present the information in Table 1 as a clear figure, and prefer to keep the information organised as a table.

*24: p22 line10 – p24 line6: This part overlaps with section 3 so make it short and combine with section 3. The key message here is that the moisture uptake region coincides with AR tracks. The right side of figure 7 is not necessary and distracting.*

From our perspective, the logic of the results on moisture sources will be less clear and potentially confusing when this section is combined with Sec. 3. Some overlap with section 3 can not be avoided in this section, and we applied some shortening where possible. We removed the right side of Fig. 7 and the corresponding text, shortening this section overall.

*25: Figure 8: the colors of the first two dashed lines are a bit difficult to distinguish. Could you change one of the colors?*

We changed the second dashed line to black.

*26: Figure 8: Could you also make figures of mean or medians of these pdfs since that is mainly discussed in the text?*

We included indicators for the mean at time 12 UTC in the revised Fig. 8.

*27: p28, line 34: If I understand correctly, Y10 uses isoRSM. Based on your discussion, can we consider the claims of Y10 are more reliable?*

We consider the approach of Y10 as more comprehensive, and therefore as potentially more reliable. However, comparison to paired vapour and precipitation measurements are in our view needed to back this expectation with evidence, as we note in the conclusions.

*28: Abstract line16-18: Make it more accessible to readers who only read the abstract.*

We rephrased the sentences to make the abstract more accessible:

"The isotope signal from the cloud level became apparent at ground level after a transition period that lasted up to several hours. Moisture source diagnostics for the periods when the cloud signal dominates show that the moisture source conditions are then partly reflected in surface precipitation and water vapour isotopes."

*29: Abstract line 20: "AR events in California"*

This paragraph was rewritten. The phrase"AR events in California" is no longer there.

*30: Abstract line 20-25: This part may not be suitable for the abstract. The conclusions are too ambiguous.*

We rephrased these conclusions, noting that in our view the results obtained in Sec. 5.1 are sufficiently robust to issue a recommendation for careful interpretation:

"In our study, the isotope signal in surface precipitation during the AR event reflects a combination of atmospheric dynamics, moisture sources and atmospheric distillation, as well as cloud microphysics and below-cloud processes. Based on this finding, we recommend careful interpretation of results obtained from Rayleigh distillation models in such events, in particular for the interpretation of surface vapour and precipitation from stratiform clouds."

**Anonymous Referee #2**

*The research presented by Weng et al. deals in detail with winter 2016 AR event in Norway. Although the study is really interesting, some parts are a bit hard to read. This is especially obvious in parts of introduction, but as well in discussion. In general, discussion part of the paper is lacking references. As well, references should be written properly and not using acronyms.*

*1. p3, line 10 – p4 line 11: This part does not belong to the Introduction and should be moved to later part of the manuscript.*

In line with the comment from reviewer 1, comment #2, this has been changed accordingly in the revised manuscript.

*2. p4 line 13-22: This part should be rewritten in more concise way.*

This section has been revised for conciseness.

*3. p4 line 28: the reference to web page should be written differently and should be listed properly in list of references.*

The reference to this website has been included as url reference.

*4. Same goes for P4 line 32 and P8 line 3.*

OK, see the comment above.

*5. p9 fig 1. I believe panel b would be sufficient since you are using ERA-Interm data. Thus, in p8 lines 4-5 are not necessary.*

We removed the satellite data and according descriptions (see reviewer 1, comment #7)

5. *p21 (Table 1 and line 3-16) it would be worthy to put observed data in the table to enable easier comparison. Doing so, this part of the text can be significantly reduced or even eliminated.*

The observed data is already part of Table 1. In our opinion, text and table work well together, depending on reader preference. We have shortened the writing substantially (see reviewer 1, comment #23).

5. *Fig 8. It is really hard to read it. Legend should be set to be representative for all panels (this way looks that is valid only for b panel). As well, please check solid and dotted lines in figure description are designated properly.*

The legend was moved to the outside of panel b, and lines are now offset for clarity. The figure caption has been corrected to be consistent with the legend.

6. *p27, line 24. Other factors*

Unfortunately, we could not understand this reviewer comment.

7. *p 28, line 7 ???*

Unfortunately, we could not understand this reviewer comment, but rephrased the paragraph slightly to improve readability.

---

## Author Response (AR2)

We are grateful for both reviewer's minor comments that help us to improve the clarity of the manuscript.

In addition to the point-by-point responses, we have carried out the following minor changes:
- have updated the uncertainties of the secondary laboratory standards (Line 5, Page 5; Line 5, Page 6).
- have updated the long-term reproducibility of liquid sample analysis at FARLAB (Line 6, Page 6).
- have acknowledged to the editor and two anonymous referees (Line 17, Page 32).
- corrected the spelling "distrometer" to "disdrometer" (Line 11, 17, Page 4; Line 4, Page 29), "paried" to "paired" (Line 13, Page 28).

The detailed point-by-point responses to all referee comments are provided below.

**Anonymous Referee #1**

*The paper is greatly improved, and I appreciate the authors' effort. I now have few minor comments.*

*1. p2, line 9: S -> "South"*

Implemented.

*2. p2, line 9: "However, an observational confirmation … remains elusive." Did you address this issue in this paper? If not, I would remove this sentence.*

Yes, we address this issue with the moisture source analysis. We think it is important to bring this up here as part of the research gap. We have slightly rephrased this sentence to "However, such model-derived estimates currently lack observational confirmation."

*3. p2, line 19: There is a recent paper by Toride et al (2021), which showed the potential of water isotope to improve weather forecasting. I think this paper should be cited in this context.*

*Toride, K., Yoshimura, K., Tada, M., Diekmann, C., Ertl, B., Khosrawi, F., & Schneider, M. (2021). Potential of mid-tropospheric water vapor isotopes to improve large-scale circulation and weather predictability. Geophysical Research Letters, 48, e2020GL091698. https://doi.org/10.1029/2020GL091698*

Have now cited the mentioned paper.

**Anonymous Referee #2**

*This new, improved version of manuscript by Weng et al. addresses all comments made by reviewers so far. The manuscript is now easier to read, with more clarified sections of the text as well as with improved figures. Only change that should be done is no to use acronyms for references which introduces confusion.*

Have now replaced the acronyms for references (i.e. C08, Y10 and C15) with the full citations (i.e. Coplen et al., 2008; Yoshimura et al., 2010 and Coplen et al., 2015). This required minor rephrasing of several sentences.